# ZeroS: Zero-Sum Linear Attention for Efficient Transformers

**Jiecheng Lu**[1†], **Xu Han**[2], **Yan Sun**[1], **Viresh Pati**[1], **Yubin Kim**[1], **Siddhartha Somani**[1], **Shihao Yang**[1‡]

Georgia Institute of Technology[1], Amazon Web Services[2]

jlu414@gatech.edu[†], shihao.yang@isye.gatech.edu[‡]

## Abstract

Linear attention methods offer Transformers $O(N)$ complexity but typically underperform standard softmax attention. We identify two fundamental limitations affecting these approaches: the restriction to convex combinations that only permits additive information blending, and uniform accumulated weight bias that dilutes attention in long contexts. We propose Zero-Sum Linear Attention (ZeroS), which addresses these limitations by removing the constant zero-order term $1/t$ and reweighting the remaining zero-sum softmax residuals. This modification creates mathematically stable weights, enabling both positive and negative values and allowing a single attention layer to perform contrastive operations. While maintaining $O(N)$ complexity, ZeroS theoretically expands the set of representable functions compared to convex combinations. Empirically, it matches or exceeds standard softmax attention across various sequence modeling benchmarks. The code implementation is available at this link.

## 1 Introduction

The Transformer architecture [1] has revolutionized sequence modeling across NLP, vision, speech, and reinforcement learning [2–7]. While its self-attention mechanism offers exceptional modeling flexibility, the quadratic $O(N^2)$ complexity in both time and memory with sequence length $N$ limits its efficient implementation to long-context scenarios [8, 9]. Researchers have developed numerous linear-time attention mechanisms [8, 10–14] that preserve Transformer's strengths while scaling to longer sequences. Approaches include sparse attention patterns [15–17], kernel methods [8, 13, 14], low-rank approximation [18, 19], and efficient factorizations [20, 21]. Despite reducing from $O(N^2)$ to $O(N)$, these variants often underperform standard softmax attention, raising the question: Why do linear approximations save computation but sacrifice accuracy? Recent efforts to bridge this gap typically: 1) hybridize linear attention with local quadratic windows [22, 23], 2) learn softmax matrix low-rank projections [19, 24], or 3) sharpen linear kernels through normalization and gating [11, 12, 25]. While offer incremental gains, these approaches often compromise $O(N)$ efficiency, rely on task-specific hyperparameters, or introduce instabilities, limiting their practical use.

In this paper, we identify two fundamental limitations affecting linear and even softmax attention: 1) Bottleneck of convex combination [26–28]: softmax attention produces convex combinations of value vectors, with linear attention also aiming to achieve this primarily for numerical stability. However, these combinations can only blend information additively, unable to express subtractive or contrastive operations directly, forcing models to use multiple layers even for simple differencing tasks. 2) Uniform weight bias and attention dilution [11, 12, 29]: In long contexts, attention mechanisms incorporate a roughly uniform $\frac{1}{N}$ component in their weight expansion, introducing a persistent averaging effect that weakens focused attention and limits modeling of complex patterns. These limitations stem from the Taylor expansion $\exp(\boldsymbol{q} \cdot \boldsymbol{k}) = 1 + \langle q, k \rangle + \frac{1}{2}\langle q, k \rangle^2 + \dots$, where the constant zero-order term enforces non-negativity for stability but creates an average-pooling bias that diminishes high-order token interactions. Rather than designing complex kernels to approximate

softmax while preserving the constant term, we propose a simpler solution: **remove it**. Subtracting the uniform component creates naturally zero-sum weights that permit both positive and negative values, enabling contrastive updates and sharper attention distributions while maintaining stability.

From this insight, we introduce ZeroS (Zero-Sum linear attention), achieving linear complexity while matching or exceeding quadratic softmax attention performance through three key elements: 1) Zero-order subtraction: removing the uniform $1/t$ term from each softmax row to create stable zero-sum weights; 2) Radial–angular decoupling: separating magnitude from direction by applying learned gates to first-order (linear) and higher-order (non-linear) softmax residuals, then reintroducing signed $\cos\theta$ terms to restore directional effects; 3) Linear-time implementation: using separable logits and gating for the reweighted zero-sum softmax, combined with linearizable angular computations via prefix sums, maintaining $O(Nd^2)$ runtime and $O(d^2)$ memory.

Our contributions include: 1) Identifying why the uniform zero-order softmax term limits attention mechanisms and demonstrating that its removal is safe and beneficial. 2) Developing Zero-Sum Linear Attention (ZeroS), a linear-time attention supporting negative weights with theoretical stability independent of sequence length. 3) Proving ZeroS offers greater expressivity than convex combinations while maintaining numerical stability. 4) Demonstrating that ZeroS matches or exceeds standard softmax attention on various benchmarks while maintaining linear time complexity.

## 2 Background

### 2.1 Preliminaries: Attention Mechanisms

We consider an input token sequence of length $N$, represented by the feature matrix $\mathbf{X} \in \mathbb{R}^{N \times d}$, where each row $\boldsymbol{x}_t \in \mathbb{R}^{1 \times d}$ is the embedding at time step $t$. With $\mathbf{Q} = \mathbf{X}\mathbf{W}_q$, $\mathbf{K} = \mathbf{X}\mathbf{W}_k$, $\mathbf{V} = \mathbf{X}\mathbf{W}_v$, an autoregressive (causal) single-head attention layer can be written in its matrix form as

$$\text{Attn}(\mathbf{X}) = \sigma\big(\mathbf{M} \odot (\mathbf{Q}\mathbf{K}^\top)\big)\,\mathbf{V}\,\mathbf{W}_o, \quad \mathbf{X} \leftarrow \mathbf{X} + \text{Attn}\big(\text{LN}(\mathbf{X})\big),$$

where $\mathbf{W}_q, \mathbf{W}_k, \mathbf{W}_v, \mathbf{W}_o \in \mathbb{R}^{d \times d}$ are learned projections, $\text{LN}(\cdot)$ denotes layer normalization, and $\mathbf{M} \in \mathbb{R}^{N \times N}$ is the causal mask with $\mathbf{M}_{ij} = 1\{i \geq j\} - \infty \cdot 1\{i < j\}$, ensuring each position attends only to itself and the past. When $\sigma$ is the row-wise softmax with a $1/\sqrt{d}$ factor, this represents standard self-attention with $O(N^2)$ complexity; replacing $\sigma$ by the linearized kernels yields the linear attention variants that can be computed in $O(N)$ [8, 14]. Omitting the causal mask $\mathbf{M}$ reverts this to encoder-only attention, attending to all pairs of positions.

**Recurrent Form** Attention admits an equivalent step-by-step formulation. At time $t$, let $\boldsymbol{q}_t = \boldsymbol{x}_t\mathbf{W}_q$, $\boldsymbol{k}_t = \boldsymbol{x}_t\mathbf{W}_k$, $\boldsymbol{v}_t = \boldsymbol{x}_t\mathbf{W}_v$. Then the output $\boldsymbol{o}_t \in \mathbb{R}^{1 \times d}$ is $\boldsymbol{o}_t = \frac{\sum_{i=1}^t \sigma(\boldsymbol{q}_t, \boldsymbol{k}_i)\,\boldsymbol{v}_i}{\sum_{i=1}^t \sigma(\boldsymbol{q}_t, \boldsymbol{k}_i)}$ where $\sigma(\boldsymbol{q}, \boldsymbol{k}) = \exp(\boldsymbol{q}\,\boldsymbol{k}^\top / \sqrt{d})$ for vanilla attention. By choosing a kernel feature map $\phi(\cdot)$ such that $\sigma(\boldsymbol{q}_t, \boldsymbol{k}_i) = \phi(\boldsymbol{q}_t)\,\phi(\boldsymbol{k}_i)^\top$, the summations can be rearranged to maintain only the $d \times d$ hidden state $\sum_{i=1}^t \phi(\boldsymbol{k}_i)^\top \boldsymbol{v}_i$, avoiding the full $N \times N$ matrix $\mathbf{Q}\mathbf{K}^\top$. This yields the linear attention formulation: $\boldsymbol{o}_t = \frac{\phi(\boldsymbol{q}_t) \sum_{i=1}^t \phi(\boldsymbol{k}_i)^\top \boldsymbol{v}_i}{\phi(\boldsymbol{q}_t) \sum_{i=1}^t \phi(\boldsymbol{k}_i)^\top}$. Replacing the summation limit $t$ with $N$ converts this from the decoder-only autoregression into a encoder-only global recurrence, summing over all positions.

### 2.2 The intuition from existing linear attention research

We begin with insights from previous research on linear attention to introduce two key elements of our ZeroS structure: 1) radial-angular decoupling, and 2) zero-sum reweighted softmax. In softmax attention, each value vector $\boldsymbol{v}_i$ is assigned a weight $\frac{\exp(\boldsymbol{q}_t\boldsymbol{k}_i)}{\sum_{i=1}^t \exp(\boldsymbol{q}_t\boldsymbol{k}_i)}$, forming a convex combination that ensures numerical stability by keeping outputs within the convex hull of $\{\boldsymbol{v}_i\}$ [26, 27, 30]. Linear attention variants attempt to approximate this using weights of linearized kernel form $\frac{\phi(\boldsymbol{q}_t)\phi(\boldsymbol{k}_i)}{\sum_{i=1}^t \phi(\boldsymbol{q}_t)\phi(\boldsymbol{k}_i)}$ [8, 11, 14]. However, without constraining the sign of $\phi(\boldsymbol{q}_t)\phi(\boldsymbol{k}_i)$, this reduces to an affine combination that lacks the stability-ensuring bounds of convexity. While researchers have addressed this using non-negative feature maps like 1+ELU and ReLU [8, 12, 31, 32], these stability-ensuring modifications still underperform compared to standard softmax attention [8, 11, 14].

**Coupling Interaction Between Radial and Angular Components** In softmax attention, the core weight term $\exp(\|\boldsymbol{q}_t\|\|\boldsymbol{k}_i\|\cos\theta)$ is controlled by both vector magnitudes and their angle $\theta$. Crucially,

when cosine flips from positive to negative, large positive values transform into very small ones, with step $t$ and $i$ highly coupled within the exponential. In contrast, linear attention applies nonlinear mappings $\phi(\cdot)$ to query and key [8, 12, 33], calculating $\|\phi(\boldsymbol{q}_t)\|\|\phi(\boldsymbol{k}_i)\| \cos \theta'$. Since these mappings yield only positive values, angles between vectors become restricted to less than 90 degrees, and the angular representation loses its flipping effect—cosine values merely serve as smooth gating signals between (0, 1). Previous research shows minimal performance changes when replacing softmax with sigmoid, ReLU, or similar functions [34–39], indicating that softmax attention's performance derives from modeling coupled angular and magnitude of $(t, i)$ pairs rather than from the exponential property itself. Therefore, when constructing linear attention, we should reimplement these complex interactions rather than attempting to approximate softmax or merely mimicking an inner product.

**Convexity of Sum-to-One Weights** Under this perspective, we revisit the convex combination in softmax attention, which primarily serves numerical stability by preserving norm regardless of sequence length. As weights become more uniform, output norm expectation decreases at approximately $1/\sqrt{t}$ with sequence length $t$ (assuming zero-mean vectors). However, these strictly positive weights mean input signals $\boldsymbol{v}_i$ can only contribute additively to outputs. In linear attention, without methods to suppress historical weights, this accumulation leads to attention dilution [11, 12, 29], where uniform signals increasingly dominate as sequence length grows. While some approaches address this using local windows or convolutional methods [9, 31, 40], these represent engineering solutions rather than resolving fundamental limitations of positive weights. Studies [27, 41–43] show that with softmax weights, a single attention layer cannot express differential or contrastive operations (even with just two tokens). The strictly positive convex combination inherently constrains ability to compress complex operations, limiting parameter efficiency. To enable more flexible parameterization with negative values, we must maintain numerical stability without relying on convex combinations' norm-preserving property while satisfying linear-time requirements.

**Flexible Weighting in Related Works** Implementing both the angular flipping effect and expressiveness requires numerically stable modeling of negative weights. Previous research [28, 44] demonstrated that negative weights improve model performance, while Differential transformer [45] showed benefits from differencing two attention matrices to obtain flexible weights. In linear attention, operations that reduce or delete historical state matrix elements outperform simple accumulation approaches [9, 14, 31, 46, 47]. In the following sections, we will show that our ZeroS method constructs zero-sum weights based on softmax, improving performance while maintaining numerical stability compared to both standard and linear attention variants. Compared to previous linear attention, ZeroS enables more effective control of radial weights and decoupled angular components in $(t, i)$ pairs from step $t$ information.

## 3 Methodology

In this section, we demonstrate that using softmax residual terms with zero-sum weights (eliminating zero-order terms) and decoupling radial-angular components in linear attention achieves three key objectives: 1) enabling numerically stable negative weights in a single attention layer for expressing differential and contrastive operations, 2) capturing the essential length-angle interactions in attention weights that allow positive-negative flipping effects, and 3) permitting the current step $t$ to effectively influence shareable accumulated weights while maintaining linear time complexity. The overall architecture of the final ZeroS block introduced in this section is shown in Fig. 1.

### 3.1 The Expansion of Softmax Function

Recent research has attempted to approximate softmax using Taylor expansions [48–51]. For input scalars $\{s_i\}_{i=1}^t$, with $\bar{s} = \frac{1}{t} \sum_{j=1}^t s_j$ and $\delta_i = s_i - \bar{s}$, the second-order Taylor expansion is:

$$\text{softmax}(s_i) \approx \frac{1}{t} + \frac{1}{t}\delta_i + \frac{1}{2t}\left(\delta_i^2 - \frac{1}{t}\sum_{j=1}^t \delta_j^2\right) + O(\|s\|^3).$$

The zero-order term $\frac{1}{t}$ ensures $\sum_i \text{softmax}(s_i) = 1$, while first-order terms reflect linear response, and higher-order terms capture nonlinear interactions and competitive relationships between weights. Computing second-order terms based on $s_{t,i} = \boldsymbol{q}_t \boldsymbol{k}_i^\top$ would require $O(d^3)$ complexity [49], making them impractical. Our approach differs: we use logits that depend only on step $i$, calculate full

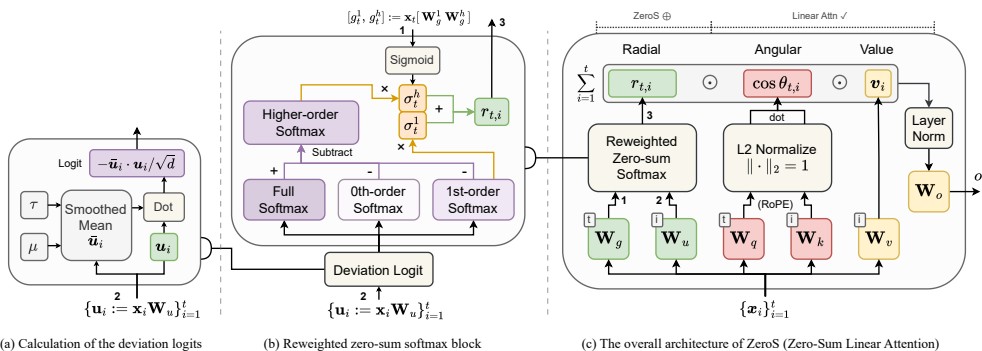

(a) Calculation of the deviation logits     (b) Reweighted zero-sum softmax block     (c) The overall architecture of ZeroS (Zero-Sum Linear Attention)

Figure 1: Illustration of the zero-sum linear attention block, including the computation of deviation logits and the reweighted zero-sum softmax operation

softmax, zero and first-order terms, derive higher-order terms through their differentiation, and employ $t$-step-dependent gating factors to achieve interaction between $(t, i)$ pairs at different orders.

The zero-order baseline primarily provides accumulated magnitude measurement, contributing $\frac{1}{\sqrt{t}}$-level norm reduction and convexity properties. However, it enables no interaction between scores. Eliminating this term creates zero-sum residual weights with both positive and negative values reflecting interaction strength. While full softmax encodes higher-order competitive effects only through positive weight magnitudes, zero-sum residual weights directly express these relationships between vectors based on the positive and negative weights, emphasizing contrastive components.

**Proposition 3.1** (Convex vs. Zero-Sum Span). *Let $\{v_i\}_{i=1}^t \subset \mathbb{R}^d$, and write $\mathcal{C} = \{\sum_i \alpha_i v_i : \alpha_i \geq 0, \sum_i \alpha_i = 1\}$, $\mathcal{Z} = \{\sum_i w_i v_i : \sum_i w_i = 0\}$, where we denote the $(t-1)$-simplex by $\Delta_{t-1} = \{\alpha \in \mathbb{R}^t : \alpha_i \geq 0, \sum_i \alpha_i = 1\}$. Then, letting $v_{\text{avg}} = \frac{1}{t}\sum_i v_i$, $\{\sum_i \alpha_i v_i - v_{\text{avg}} : \alpha \in \Delta_{t-1}\} \subsetneq \{\sum_i w_i v_i : \sum_i w_i = 0\}$, i.e. the zero-sum span of $\{v_i - v_{\text{avg}}\}$ strictly contains the deviations achievable by convex weights, with strictness whenever the $v_i$ are not all identical.*

**Corollary 3.2** (Expressive Gain of Zero-Sum Attention). *In a residual block $x_t \mapsto x_t + \sum_i w_i v_i$, softmax weights $w_i = \alpha_i$ yield head deviations in $\{\sum_i \alpha_i v_i - v_{\text{avg}} : \alpha \in \Delta_{t-1}\}$; zero-order subtraction $w_i = \alpha_i - \frac{1}{t}$ yields head deviations in $\{\sum_i w_i v_i : \sum_i w_i = 0\}$. Since $\{\sum_i \alpha_i v_i - v_{\text{avg}}\} \subsetneq \{\sum_i w_i v_i : \sum_i w_i = 0\}$, zero-sum attention enlarges the set of deviation vectors the head can produce (and hence its expressivity), only without the uniform average direction.*

Zero-sum weights can express more complex interactions after removing the zero-order term, with expressivity reduction only in the orthogonal direction of $v_{\text{avg}}$. This direction typically represents the lowest-cost basis since it requires only average pooling. We can recover this capability through multiple attention heads and layer stacking. To strictly ensure this direction is not lost, our implementation retains the zero-order term in the first layer, removing it in subsequent layers as described below.

## 3.2 Reweighted Zero-sum Softmax

We define the reweighted zero-sum softmax operation. For logit input $s_{t,i}$ at step $t$, we compute: $\bar{s}_t = \frac{1}{t}\sum_{j=1}^t s_{t,j}$, $\delta_{t,i} = s_{t,i} - \bar{s}_t$. Subtracting zero-order $(1/t)$ and first-order $(\delta_{t,i}/t)$ terms from softmax yields the residual:

$$\varepsilon_{t,i} = \frac{\exp(s_{t,i})}{\sum_{i=1}^t \exp(s_{t,i})} - \frac{1}{t} - \frac{\delta_{t,i}}{t} = O(\|\delta\|^2),$$

where $\sum_i \varepsilon_{t,i} = 0$ and $\sum_i \delta_{t,i}/t = 0$. We gate these components using learned scalars $\sigma_t^1 = \text{sigmoid}(g_t^1)$ and $\sigma_t^h = \text{sigmoid}(g_t^h)$, defining zero-sum weights:

$$w_{t,i} = \sigma_t^1 \frac{\delta_{t,i}}{t} + \sigma_t^h \varepsilon_{t,i}, \quad \sum_{i=1}^t w_{t,i} = 0.$$

This form assigns two gating weights: one for the first-order orthogonal direction and another for all directions of second-order and above. For the first attention layer, we can optionally preserve

the zero-order term using $\sigma_t^0 = \tanh(g_t^0)$, giving $w_{t,i}' = \sigma_t^0 \frac{1}{t} + \sigma_t^1 \frac{\delta_{t,i}}{t} + \sigma_t^h \varepsilon_{t,i}$, though experiments show this has minimal impact across most tasks.

**Remark.** A key advantage of this formulation for linear-time attention is that even with logits $s_{t,i} = s_i$ that are independent of $t$, we can still control interactions of different orders in the final weights $w_{t,i}$ through the $t$-step gating mechanism $\sigma_t^{(\cdot)}$ across orthogonal directions from softmax expansion. This gate reweighting approach $w_{t,i} = \sigma_t^1 \frac{\delta_i}{t} + \sigma_t^h \varepsilon_i$ effectively replaces the traditional linearization that decomposes $\exp(\boldsymbol{q}_t \boldsymbol{k}_i)$ into $\phi(\boldsymbol{q}_t)\phi(\boldsymbol{k}_i)$.

**Proposition 3.3** (Preservation of Affine Hull and Expressivity). *Let $\{\boldsymbol{v}_i\}_{i=1}^t \subset \mathbb{R}^d$ and write $\boldsymbol{v}_{\text{avg}} = \frac{1}{t}\sum_{i=1}^t \boldsymbol{v}_i$, $\Delta_i = \boldsymbol{v}_i - \boldsymbol{v}_{\text{avg}}$. A single head with full softmax (or full reweighted softmax with the zero-order term kept) can produce any point in the affine hull*

$$\text{Aff}\{\boldsymbol{v}_1, \ldots, \boldsymbol{v}_t\} \;=\; \Big\{\boldsymbol{v}_{\text{avg}} + \sum_{i=1}^t \alpha_i \,\Delta_i : \sum_{i=1}^t \alpha_i = 1\Big\}.$$

*A single head without the zero-order term (i.e. zero-sum weights) can produce any point in the linear span*

$$\text{Span}\{\Delta_1, \ldots, \Delta_t\} \;=\; \Big\{\sum_{i=1}^t w_i \,\Delta_i : \sum_{i=1}^t w_i = 0\Big\}.$$

*Therefore, if you use one head (or one layer) that retains the zero-order term and then stack one or more heads (layers) that subtract it, the Minkowski sum of their reachable sets is exactly*

$$\text{Aff}\{\boldsymbol{v}_i\} \;+\; \text{Span}\{\Delta_i\} \;=\; \text{Aff}\{\boldsymbol{v}_i\}.$$

In other words, after the first full attention layer, it already cover the entire affine hull, and the subsequent zero-sum attentions do not shrink that. The overall network can still express any affine combination of the $\boldsymbol{v}_i$.

**Residual Stream Alignment** When value vectors $\{v_i\}$ are centered and i.i.d., zero-sum attention produces $o_t = \sum_{i=1}^t w_{t,i} v_i$ with $\sum_i w_{t,i} = 0$ and $\mathbb{E}[o_t] = 0$. This aligns with decoder-only Transformer's residual stream ideology where $x_t \leftarrow x_t + \text{Attn}(x_t)$ should provide pure updates without constant bias. Subtracting the zero-order term naturally centers these residuals, improving training stability.

We now show that the reweighted zero-sum softmax achieves the same level of numerical stability as the original softmax.

**Lemma 3.4** (Numerical Stability of Zero-Sum Softmax). *Let $w_{t,i}$ be the reweighted zero-sum softmax weights with $\sum_i w_{t,i} = 0$, and assume each value vector satisfies $\|v_i\| \leq B$. Then for any step $t$,*

$$\Big\|\sum_{i=1}^t w_{t,i} v_i\Big\| \;\leq\; \max_i |w_{t,i}| \sum_{i=1}^t \|v_i\| \;\leq\; B\, t\, \max_i |w_{t,i}|.$$

*Moreover, since $|w_{t,i}| \leq \max\big(\frac{1}{t}, \frac{|\delta_{t,i}|}{t}, \frac{|\rho_{t,i}|}{2t}\big)$ and $\delta, \rho = O(1)$ under bounded logits, we have $\max_i |w_{t,i}| = O(1/t)$ and hence $\big\|\sum_i w_{t,i} v_i\big\| = O(B)$, independent of $t$.*

With controllable logits generation methods independent of $t$, such as the scaled dot-product in original softmax attention, the numerical stability of the above method remains well-controlled. This allows us to employ zero-sum weights that permit negative values while still achieving numerically stable outputs, even without the norm-preserving property of convex combinations.

**Proposition 3.5** (Uniform Lipschitz Bound of Zero-Sum Softmax with decay factor $1/\sqrt{t}$). *Assume each value vector satisfies $\|\boldsymbol{v}_i\| \leq B$, each pre-softmax logit $s_{t,i}(\boldsymbol{x}) \in [-S, S]$ is $L_s$-Lipschitz in the input $\boldsymbol{x}$, each residual weight $w_{t,i}(\boldsymbol{x})$ obeys the scaling $|w_{t,i}(\boldsymbol{x}) - w_{t,i}(\boldsymbol{x}')| \leq \frac{L_w}{t}\|\boldsymbol{x} - \boldsymbol{x}'\|$, for some constant $L_w$ depending only on $S$ and the sigmoid gates. Let the head output be $\boldsymbol{o}_t(\boldsymbol{x}) = \frac{1}{\sqrt{t}}\sum_{i=1}^t w_{t,i}(\boldsymbol{x})\,\boldsymbol{v}_i$, $\sum_{i=1}^t w_{t,i}(\boldsymbol{x}) = 0$. Then for any two inputs $\boldsymbol{x}, \boldsymbol{x}'$,*

$$\big\|\boldsymbol{o}_t(\boldsymbol{x}) - \boldsymbol{o}_t(\boldsymbol{x}')\big\| \;\leq\; \frac{B\, L_w}{\sqrt{t}}\,\|\boldsymbol{x} - \boldsymbol{x}'\|.$$

*The zero-sum update is $\ell_2$-Lipschitz in its inputs with constant $O(1/\sqrt{t})$, ensuring stable gradients and activations independent of sequence length.*

This proposition introduces a $1/\sqrt{t}$ decay factor that ensures reweighted zero-sum softmax maintains variance reduction similar to convex combinations, promoting training stability. However, since linear attention methods typically apply Layer Normalization to control output variance, LayerNorm effectively supersedes this factor and is sufficient to ensure gradient stability during training.

**Reweighted Zero-sum Softmax in Linear Time** To achieve linear-time computation, we simplify logits from $s_{t,i}$ to $s_i$ by removing t-dependency. While we could use basic forms like $s_i = \boldsymbol{x}_i \mathbf{W}_s^{d\times 1}$ or quadratic forms $s_i = \boldsymbol{x}_i \mathbf{W}_s \mathbf{W}_s^\top \boldsymbol{x}_i^\top / d$ to emulate dot-products, we instead propose a design with a more meaningful representation.

We want these logits to express the deviation of step $i$ relative to previous steps. We calculate the negative inner product between each step's vector $\boldsymbol{u}_i = \boldsymbol{x}_i \mathbf{W}_u$ and its cumulative average. For better assessment of initial steps, we introduce trainable parameters $\mu \in \mathbb{R}^{1\times d}$ and $\tau \in \mathbb{R}$ as a smoothing prior, calculating deviation logits $s_i$ as:

$$s_i = -\frac{1}{\sqrt{d}} \boldsymbol{u}_i \bar{\boldsymbol{u}}_i^\top, \text{ where } \bar{\boldsymbol{u}}_i = \frac{e^\tau \mu + \sum_{j=1}^i \boldsymbol{u}_j}{e^\tau + i}$$

Let $r_{t,i}$ represent the final computed reweighted softmax result:

$$\delta_{t,i} = s_i - \bar{s}_t, \; \varepsilon_{t,i} = \frac{\exp(s_i)}{\sum_{i=1}^t \exp(s_i)} - \frac{1}{t} - \frac{\delta_{t,i}}{t},$$

$$r_{t,i} = \sigma_t^1 \frac{\delta_{t,i}}{t} + \sigma_t^h \varepsilon_{t,i} = \frac{\sigma_t^h}{\sum_{i=1}^t \exp(s_i)} \exp(s_i) + \frac{(\sigma_t^1 - \sigma_t^h)}{t}(s_i - \bar{s}_t) - \frac{\sigma_t^h}{t}$$

where $\sigma_t^1 = \text{sigmoid}(\boldsymbol{x}_t \mathbf{W}_g^1)$, $\sigma_t^h = \text{sigmoid}(\boldsymbol{x}_t \mathbf{W}_g^h)$, $\mathbf{W}_g^1, \mathbf{W}_g^h \in \mathbb{R}^{d\times 1}$. The terms dependent on $t$ and $i$ are effectively separated, enabling linear-time computation through prefix sums.

### 3.3 ZeroS Linear Attention: Interaction Between Radial and Angular Components

The reweighted zero-sum softmax provides strong foundations for linear attention by yielding numerically stable weights (including negative values) with computational simplicity while enabling high-order $(t, i)$ interactions in token mixing. We leverage linear-time logit inputs that depend only on step $i$ and implement effects on different softmax orders through step $t$ gating.

However, our earlier discussion showed that the angle-flipping effect in softmax attention's $\exp(\|\boldsymbol{q}_t\|\|\boldsymbol{k}_i\|\cos\theta)$ significantly impacts final weights. While reweighted zero-sum softmax effectively models length interactions through $i$-step logits and $t$-step gating, it lacks control over directional influence when measuring vector differences in $(t, i)$ pairs. Since zero sum weights provide inherent stability, no longer need to place $\cos\theta$ in the denominator normalizer, we can directly multiply the angular component ($\cos\theta$) with the reweighted softmax radial component without positivity constraints. This approach enables seamless integration with rotary positional embedding (RoPE) [52], making the angle term's role in measuring relative distance more explicit.

**Zero-Sum Linear Attention (ZeroS)** We use normalized vectors $\hat{\boldsymbol{k}}_i = \boldsymbol{k}_i/\|\boldsymbol{k}_i\|$ and $\hat{\boldsymbol{q}}_t = \boldsymbol{q}_t/\|\boldsymbol{q}_t\|$, with $r_{t,i}$ as the radial component and $\cos\theta$ as the angular component. ZeroS produces the output:

$$\boldsymbol{o}_t = \sum_{i=1}^t r_{t,i} \cos\theta \, \boldsymbol{v}_i, \quad \cos\theta = \hat{\boldsymbol{q}}_t \hat{\boldsymbol{k}}_i^\top$$

With RoPE's block-diagonal rotary matrix applied, the angular term becomes $\cos\theta' = \hat{\boldsymbol{q}}_t \mathbf{R}_{t-i} \hat{\boldsymbol{k}}_i^\top$. Both $r_{t,i}$ and $\cos\theta$ are centered values, preserving zero-sum properties in the weights $r_{t,i}\cos\theta$. Though not strictly positive (unlike traditional radial components), $r_{t,i}$ captures magnitude effects from step $i$, reflecting length-related interactions between $(t, i)$ pairs.

**Linear-Time Scan** With logits that depend only on step $i$ (e.g. $s_i = -\frac{1}{\sqrt{d}} \boldsymbol{u}_i \bar{\boldsymbol{u}}_i^\top$ with $\boldsymbol{u}_i = \boldsymbol{x}_i \mathbf{W}_u$ and $\bar{\boldsymbol{u}}_i = \frac{e^\tau \mu + \sum_{j=1}^i \boldsymbol{u}_j}{e^\tau + i}$), the radial weight at time $t$ can be decomposed into the full softmax term, the 0th-order baseline, and the 1st-order term: $\underbrace{\frac{e^{s_i}}{\sum_{j=1}^t e^{s_j}}}_{\text{Full}}, \; \underbrace{\frac{1}{t}}_{\text{0th}}, \; \underbrace{\frac{1}{t}s_i - \frac{1}{t^2}\sum_{j=1}^t s_j}_{\text{1st}}$. Hence the higher-order zero-sum residual is Full $-$ 0th $-$ 1st. We realize ZeroS by gating the first-order zero-sum and

Table 1: Evaluation Results of ZeroS on the MAD benchmark.

| Model | Compress | Fuzzy Recall | In-Context Recall | Memorize | Noisy Recall | Selective Copy | Average |
|---|---|---|---|---|---|---|---|
| Hyena | 45.2 | 7.90 | 81.7 | 89.5 | 78.8 | 93.1 | 66.0 |
| MultiHead Hyena | 44.8 | 14.4 | 99.0 | 89.4 | 98.6 | 93.0 | 73.2 |
| Mamba | 52.7 | 6.70 | 90.4 | 89.5 | 90.1 | 86.3 | 69.3 |
| GLA | 38.8 | 6.90 | 80.8 | 63.3 | 81.6 | 88.6 | 60.0 |
| DeltaNet | 42.2 | 35.7 | 100 | 52.8 | 100 | 100 | 71.8 |
| LinAttn | 31.1 | 8.15 | 91.0 | 74.9 | 75.6 | 93.1 | 62.3 |
| Transformer | 51.6 | 29.8 | 94.1 | 85.2 | 86.8 | 99.6 | 74.5 |
| **ZeroS** | 44.0 | 14.9 | 99.9 | 88.1 | 96.1 | 97.8 | 73.5 |
| **ZeroS-SM** | 45.2 | 28.0 | 100 | 84.3 | 96.6 | 98.5 | 75.4 |

the higher-order residual with $\sigma_t^1 = \text{sigmoid}(\boldsymbol{x}_t \mathbf{W}_g^1)$ and $\sigma_t^h = \text{sigmoid}(\boldsymbol{x}_t \mathbf{W}_g^h)$, and optionally in the first layer retaining the 0th-order baseline $\sigma_t^0 = \text{sigmoid}(\boldsymbol{x}_t \mathbf{W}_g^0)$ (with fixed $\sigma_t^0 = 0$ by default). Using normalized directions $\hat{\boldsymbol{q}}_t = \boldsymbol{q}_t / \|\boldsymbol{q}_t\|$ and $\hat{\boldsymbol{k}}_i = \boldsymbol{k}_i / \|\boldsymbol{k}_i\|$, we maintain the following prefix scans at step $t$:

$$E_t = \sum_{i=1}^t e^{s_i}, \quad P_t = \sum_{i=1}^t s_i, \quad \mathbf{F}_t = \sum_{i=1}^t e^{s_i} \hat{\boldsymbol{k}}_i^\top \boldsymbol{v}_i, \quad \mathbf{G}_t = \sum_{i=1}^t s_i \hat{\boldsymbol{k}}_i^\top \boldsymbol{v}_i, \quad \mathbf{H}_t = \sum_{i=1}^t \hat{\boldsymbol{k}}_i^\top \boldsymbol{v}_i.$$

The output is then a gated activation of these scans by the current step's angular vector $\hat{\boldsymbol{q}}_t$:

$$\boldsymbol{o}_t = \hat{\boldsymbol{q}}_t \left[ \underbrace{\sigma_t^h \left( \frac{1}{E_t}\mathbf{F}_t - \frac{1}{t}\mathbf{G}_t + \left(\frac{P_t}{t^2} - \frac{1}{t}\right)\mathbf{H}_t \right)}_{\sigma_t^h \text{ (Full} - \text{0th} - \text{1st)}} + \underbrace{\sigma_t^1 \left( \frac{1}{t}\mathbf{G}_t - \frac{P_t}{t^2}\mathbf{H}_t \right)}_{\sigma_t^1 \text{ (1st restore)}} + \underbrace{\sigma_t^0 \frac{1}{t}\mathbf{H}_t}_{\sigma_t^0 \text{ (optional 0th restore)}} \right]$$

$$= \hat{\boldsymbol{q}}_t \big( \alpha_t \mathbf{F}_t + \beta_t \mathbf{G}_t + \gamma_t \mathbf{H}_t \big),$$

where

$$\alpha_t = \frac{\sigma_t^h}{E_t}, \qquad \beta_t = \frac{\sigma_t^1 - \sigma_t^h}{t}, \qquad \gamma_t = \left(\frac{P_t}{t^2} - \frac{1}{t}\right)\sigma_t^h - \frac{P_t}{t^2}\sigma_t^1 + \frac{\sigma_t^0}{t}.$$

This scan keeps only $O(d^2)$ state $(\mathbf{F}_t, \mathbf{G}_t, \mathbf{H}_t)$ and updates in $O(d^2)$ per step, yielding overall $O(Nd^2)$ time and $O(d^2)$ memory while implementing the zero-sum weighting. Moreover, our reweighted zero-sum approach can also be directly applied to standard softmax attention. See section A.1.6 for more details.

## 4  Experiments

ZeroS's zero-sum formulation enhances the attention layer's expressivity for complex operations, particularly evident in in-context learning tasks [37, 53]. We evaluate both linear-time ZeroS and quadratic-time ZeroS-SM on recent in-context learning benchmarks, along with experiments on NLP, image, and time series tasks. In all experiments, we directly replaced the multi-head attention module with ZeroS under original benchmark settings, preserving all other components (MLP/GLU, embeddings, hyperparameters) to ensure strict alignment with previous standards.

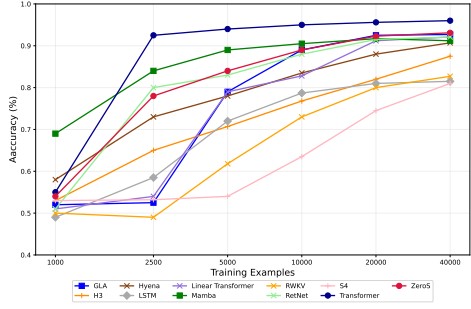

Figure 2: Evaluation of ZeroS on RegBench.

We previously described the prefix-sum computation of autoregressive ZeroS. For the encoder-only ZeroS, the summation simply spans all timesteps. We use the causal version of ZeroS for all datasets except image modeling. We provide a more detailed description of the experimental datasets in Appendix A.5. For simplicity, we do not apply first-layer 0-th order term addition in our experiments.

**MAD**  We evaluate ZeroS on the MAD benchmark [54], which tests sequence models on in-context tasks. As shown in Table 1, ZeroS outperforms other linear-time models (Hyena, Mamba, GLA, DeltaNet, LinAttn [9, 14, 31, 55]), achieving performance closest to Transformer, while ZeroS-SM further improves upon Transformer's average score. Task-level analysis shows ZeroS significantly outperforms LinAttn on In-Context and Noisy Recall tasks, supporting our hypothesis that zero-sum weights enhance algorithmic abilities. However, on tasks like Compress and Memorize that rely less on complex representations, ZeroS provides minimal gains. Unlike DeltaNet, which actively deletes memory states, ZeroS maintains strong memorization despite using negative weights, indicating that our zero-order modifications preserve sequence memory capacity.

Table 2: Evaluation Results on WikiText

| Model | PPL (val) | PPL (test) | Params (M) |
|---|---|---|---|
| FLASH | 25.92 | 26.7 | 42.17 |
| 1+elu | 27.44 | 28.05 | 44.65 |
| Performer | 62.5 | 63.16 | 44.65 |
| cosFormer | 26.53 | 27.06 | 44.65 |
| Syn(D) | 31.31 | 32.43 | 46.75 |
| Syn(R) | 33.68 | 34.78 | 46.75 |
| gMLP | 28.08 | 29.13 | 47.83 |
| S4 | 38.34 | 39.66 | 45.69 |
| DSS | 39.39 | 41.07 | 45.63 |
| GSS | 29.61 | 30.74 | 43.84 |
| RWKV-4 | 24.31 | 25.07 | 46.23 |
| LRU | 29.86 | 31.12 | 46.75 |
| TNN | 23.98 | 24.67 | 48.66 |
| Mamba | 22.58 | 23.19 | 44.99 |
| HGRN2 | 23.1 | 23.73 | 44.66 |
| Transformer | 24.4 | 24.78 | 44.65 |
| **ZeroS** | 23.91 | 24.61 | 46.31 |
| **ZeroS-SM** | 23.62 | 24.17 | 44.69 |

**MQAR**  We follow the setup of [56] for the MQAR task, which evaluates models' ability to learn induction heads for in-context associative recall. Using the same hyperparameter sweep, Fig. 3 shows ZeroS performs comparably to vanilla attention across most configurations.

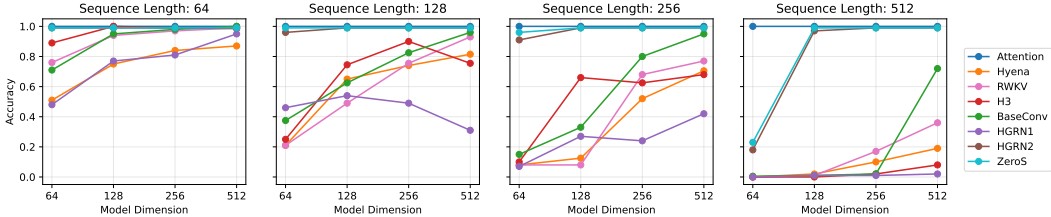

Figure 3: Performance evaluation on the MQAR benchmark, illustrating the relationship between model dimension (x-axis) and accuracy (y-axis). ZeroS demonstrates consistent performance advantages over other structures across all experimental configurations.

**RegBench**  We evaluate ZeroS on Reg-Bench [57] following the original experimental setup (Figure 2). RegBench tests models' ability to infer regular language structures from examples. ZeroS outperforms linear-time baselines including GLA, RetNet [58], and RWKV [59].

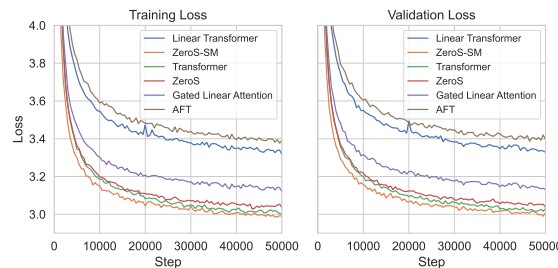

### 4.1 Language Modeling

**WikiText**  We conduct language modeling on WikiText-103 following [60]'s setup, with results in Table 2. ZeroS outperforms vanilla Transformer at this smaller scale, demonstrating its efficiency. ZeroS-SM yields further improvements, showing enhanced reasoning capability from the zero-order term removal.

Figure 4: Performance Evaluation of ZeroS on OWT2

**OWT2**  We evaluate ZeroS on OpenWebText2 (OWT2) [61] using a 12-layer, 768-dimensional GPT-2 architecture with various token layers (see Appendix A.5). Figure 4 shows ZeroS tracks much closer to vanilla Transformer than other linear methods like AFT [62] and GLA, while ZeroS-SM further improves upon vanilla Transformer performance.

**Image Modeling**  Following HGRN2 [60], wee valuate ZeroS on ImageNet by replacing the DeiT-Tiny architecture's softmax atten-

Table 3: Comparative analysis of image classification performance on ImageNet-1k.

| Model | DeiT-Tiny Top-1 Acc | Params (M) |
|---|---|---|
| DeiT | 72.20 | 5.7 |
| TNN | 72.29 | 6.4 |
| HGRN1 | 74.40 | 6.1 |
| HGRN2 | 75.39 | 6.1 |
| **ZeroS** | 75.51 | 6.0 |

Table 4: Evaluation of the ZeroS performance on the Time Series Forecasting Benchmark

| Models | ZeroS | | GLA | | AFT | | iTransformer | | PatchTST | | DLinear | |
|---|---|---|---|---|---|---|---|---|---|---|---|---|
| Metric | MSE | MAE | MSE | MAE | MSE | MAE | MSE | MAE | MSE | MAE | MSE | MAE |
| Weather | 0.218 | 0.265 | 0.223 | 0.267 | 0.220 | 0.266 | 0.232 | 0.274 | 0.221 | 0.261 | 0.233 | 0.282 |
| Solar | 0.192 | 0.256 | 0.204 | 0.266 | 0.198 | 0.259 | 0.219 | 0.284 | 0.202 | 0.254 | 0.216 | 0.277 |
| ETTh1 | 0.414 | 0.433 | 0.418 | 0.439 | 0.409 | 0.433 | 0.454 | 0.467 | 0.413 | 0.431 | 0.422 | 0.436 |
| ETTh2 | 0.341 | 0.392 | 0.342 | 0.390 | 0.337 | 0.390 | 0.374 | 0.410 | 0.330 | 0.379 | 0.426 | 0.444 |
| ETTm1 | 0.347 | 0.387 | 0.357 | 0.394 | 0.348 | 0.386 | 0.373 | 0.401 | 0.346 | 0.380 | 0.347 | 0.376 |
| ETTm2 | 0.245 | 0.312 | 0.250 | 0.315 | 0.246 | 0.311 | 0.265 | 0.332 | 0.247 | 0.312 | 0.252 | 0.326 |

Table 6: Ablation Study on the MAD benchmark.

| Model | Compress | Fuzzy Recall | In-Context Recall | Memorize | Noisy Recall | Selective Copy | Average |
|---|---|---|---|---|---|---|---|
| ZeroS | 44.0 | 14.9 | 99.9 | 88.1 | 96.1 | 97.8 | 73.5 |
| ZeroS w/ 0-th | 42.0 | 10.5 | 91.4 | 85.2 | 90.0 | 97.1 | 69.4 |
| ZeroS w/o RWSM | 36.3 | 10.6 | 91.8 | 81.7 | 89.7 | 95.3 | 67.6 |
| ZeroS w/o Gating | 39.7 | 13.5 | 96.3 | 83.0 | 94.6 | 97.8 | 70.8 |
| ZeroS w/o Norm | 39.1 | 12.3 | 89.0 | 87.0 | 91.7 | 97.1 | 69.4 |
| ZeroS-SM | 45.2 | 28.0 | 100 | 84.3 | 96.6 | 98.5 | 75.4 |

tion with our encoder-only implementation. As shown in Table 3, ZeroS outperforms previous 294 methods including TNN [63] and HGRN1 [64] under comparable parameter budgets.

**Time Series** Following the setup in [65], we evaluate ZeroS on time series forecasting tasks. ZeroS outperforms both efficient sequence models (GLA, AFT) and domain-specific approaches (iTransformer [66], PatchTST [67]) on most datasets.

## 4.2 Ablation Studies

We conduct ablation studies on MAD and WikiText-103 to analyze key components of ZeroS. Reintroducing the 0-th order softmax term reduces performance on In-Context Recall, Noisy Recall, and WikiText, confirming the representational advantage of zero-sum weights. Replacing the reweighted zero-sum softmax

Table 5: Ablation Study on WikiText-103

| Model | PPL (val) | PPL (test) | Params (M) |
|---|---|---|---|
| ZeroS | 23.91 | 24.61 | 46.31 |
| ZeroS w/ 0-th | 24.05 | 24.74 | 46.31 |
| ZeroS w/o RWSM | 24.21 | 24.97 | 46.31 |
| ZeroS-SM | 23.62 | 24.17 | 44.69 |

with standard softmax further degrades performance, highlighting the expressive gap between convex combinations and our flexible zero-sum mechanism. Ablating the gating component causes moderate performance drops across most tasks, suggesting it contributes broadly to model flexibility. Finally, removing LayerNorm notably impacts performance on In-Context Recall but not on simpler tasks like Memorize, indicating stable variance is particularly critical for algorithmic reasoning: consistent with normalization's role in linear attention mechanisms. See §A.3 for additional baselines and ablations.

## 5 Conclusion and Limitation

We introduced Zero-Sum Linear Attention (ZeroS), addressing fundamental limitations of linear attention by removing the constant zero-order term from softmax and reweighting the resulting zero-sum residuals. Our approach enables higher-order token interactions while maintaining O(N) complexity, bridging the performance gap between linear and quadratic attention methods. Evaluations across diverse tasks show ZeroS matches or exceeds standard softmax attention while offering significant efficiency advantages, challenging the belief that expressivity-efficiency tradeoffs in attention mechanisms are inevitable.

As for the limitation, our research prioritizes improving attention's algorithmic expressivity rather than providing engineering optimizations like GPU acceleration implementations found in Mamba or GLA [9, 14]. Also, our resource constraints prevented large-scale model training and evaluation on LLM benchmarks, which would involve numerous factors. This focused approach allowed us to precisely identify ZeroS's algorithmic improvements without requiring extensive engineering or computational resources that are typically needed for optimizing large benchmark metrics. Additionally, our evaluation of ZeroS primarily focuses on autoregressive tasks. Future work may explore its capabilities on non-causal tasks to further extend its applicability.

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

# A Technical Appendices and Supplementary Material

## A.1 Additional Theoretical Discussion

**Proposition A.1** (Convex vs. Zero-sum Span). *Let $v_1, \ldots, v_t \in \mathbb{R}^d$ and denote their centroid by $v_{\mathrm{avg}} = \frac{1}{t} \sum_{i=1}^{t} v_i$. Write the deviation matrix*

$$\Delta V = [\, v_1 - v_{\mathrm{avg}}, \, \ldots, \, v_t - v_{\mathrm{avg}} \,] \in \mathbb{R}^{d \times t}.$$

*Define the convex hull and zero-sum spaces*

$$\mathcal{C} := \Big\{ \sum_{i=1}^{t} \alpha_i v_i : \alpha_i \geq 0, \, \sum_{i=1}^{t} \alpha_i = 1 \Big\}, \qquad \mathcal{Z} := \Big\{ \sum_{i=1}^{t} w_i v_i : \sum_{i=1}^{t} w_i = 0 \Big\}.$$

*Then*

$$\mathcal{C} = v_{\mathrm{avg}} + \big\{ \Delta V \, \alpha : \alpha \in \Delta_{t-1} \big\} \subsetneq v_{\mathrm{avg}} + \big\{ \Delta V \, w : w \in \mathbb{R}^t, \, \mathbf{1}^\top w = 0 \big\} = v_{\mathrm{avg}} + \mathcal{Z},$$

*where $\Delta_{t-1}$ is the $(t-1)$-dimensional probability simplex.*

**Corollary A.2** (Expressive Capacity). *Assume the $v_i$ are affinely independent and $d \geq t - 1$, so that $\operatorname{rank} \Delta V = t - 1$. Then*

$$\dim(\mathcal{C}) = t - 1, \qquad \dim(\mathcal{Z}) = t - 1,$$

*but $\mathcal{C}$ is bounded, whereas $\mathcal{Z}$ is an unbounded linear subspace of the same dimension. Consequently $\mathcal{C} \subsetneq v_{\mathrm{avg}} + \mathcal{Z}$, and zero-sum attention can realise outputs unattainable by any convex-combination attention.*

*Proof sketch of Proposition A.1 and Corollary A.2.*

1. **Convex representation.** For any $\alpha \in \Delta_{t-1}$, $\sum_i \alpha_i v_i = v_{\mathrm{avg}} + \Delta V \, \alpha$.

2. **Zero-sum representation.** If $w \in \mathbb{R}^t$ satisfies $\mathbf{1}^\top w = 0$, then $\sum_i w_i v_i = \Delta V \, w$.

3. **Strict inclusion.** The vector $w = (1, -1, 0, \ldots, 0)$ lies in the hyperplane $\mathbf{1}^\top w = 0$ but not in the simplex $\Delta_{t-1}$; hence $v_{\mathrm{avg}} + \Delta V \, w \in v_{\mathrm{avg}} + \mathcal{Z} \setminus \mathcal{C}$.

4. **Dimensionality.** Under affine independence and $d \geq t - 1$, the matrix $\Delta V$ has rank $t - 1$. Linear images preserve dimension, giving the stated dimensions of $\mathcal{C}$ and $\mathcal{Z}$ and proving the corollary.

$\square$

### A.1.1 Proof of Proposition 3.1 (Convex vs. Zero-Sum Span).

Let $\{\boldsymbol{v}_i\}_{i=1}^{t} \subset \mathbb{R}^d$ and define

$$\mathcal{C} = \Big\{ \sum_{i=1}^{t} \alpha_i \boldsymbol{v}_i : \alpha_i \geq 0, \, \sum_{i=1}^{t} \alpha_i = 1 \Big\}, \quad \mathcal{Z} = \Big\{ \sum_{i=1}^{t} w_i \boldsymbol{v}_i : \sum_{i=1}^{t} w_i = 0 \Big\},$$

and $\boldsymbol{v}_{\mathrm{avg}} = \frac{1}{t} \sum_{i=1}^{t} \boldsymbol{v}_i$. For any $\boldsymbol{y} \in \mathcal{C}$ with weights $\alpha \in \Delta_{t-1}$,

$$\boldsymbol{y} - \boldsymbol{v}_{\mathrm{avg}} = \sum_{i=1}^{t} \Big( \alpha_i - \tfrac{1}{t} \Big) \boldsymbol{v}_i, \qquad \text{with} \quad \sum_{i=1}^{t} \Big( \alpha_i - \tfrac{1}{t} \Big) = 0.$$

Hence the centered convex set equals

$$\mathcal{C}_{\mathrm{dev}} = \Big\{ \sum_{i=1}^{t} w_i \boldsymbol{v}_i : \sum_{i=1}^{t} w_i = 0, \, w_i \geq -\tfrac{1}{t} \Big\},$$

which is $\mathcal{Z}$ with extra lower bounds on the coefficients. Therefore $\mathcal{C}_{\mathrm{dev}} \subseteq \mathcal{Z}$.

To see the inclusion is strict unless all $\boldsymbol{v}_i$ are identical, pick $j \neq k$ with $\boldsymbol{v}_j \neq \boldsymbol{v}_k$ and consider $\boldsymbol{v}_j - \boldsymbol{v}_k \in \mathcal{Z}$ (take $w_j = 1, w_k = -1$, others 0). If $\boldsymbol{v}_j - \boldsymbol{v}_k \in \mathcal{C}_{\mathrm{dev}}$, we would have

$$\alpha_j - \tfrac{1}{t} = 1, \quad \alpha_k - \tfrac{1}{t} = -1, \quad \alpha_i - \tfrac{1}{t} = 0 \, (i \notin \{j, k\}),$$

implying $\alpha_k = -1 + \tfrac{1}{t} < 0$ for $t \geq 2$, a contradiction. Thus $\boldsymbol{v}_j - \boldsymbol{v}_k \notin \mathcal{C}_{\mathrm{dev}}$, and $\mathcal{C}_{\mathrm{dev}} \subsetneq \mathcal{Z}$ whenever the $\boldsymbol{v}_i$ are not all equal. If all $\boldsymbol{v}_i$ coincide, both sets reduce to $\{0\}$. $\square$

### A.1.2 Proof of Corollary 3.2 (Expressive Gain of Zero-Sum Attention).

In a residual head $\boldsymbol{x}_t \mapsto \boldsymbol{x}_t + \sum_i w_i \boldsymbol{v}_i$, the deviation (relative to the average direction) produced by softmax weights $\alpha$ is

$$\sum_i \alpha_i \boldsymbol{v}_i - \boldsymbol{v}_{\mathrm{avg}} \in \mathcal{C}_{\mathrm{dev}}.$$

Subtracting the zero-order term corresponds to $w_i = \alpha_i - \frac{1}{t}$ with $\sum_i w_i = 0$, hence deviations lie in $\mathcal{Z}$. By Proposition 3.1, $\mathcal{C}_{\mathrm{dev}} \subsetneq \mathcal{Z}$ (for non-degenerate $\{\boldsymbol{v}_i\}$), so zero-sum attention strictly enlarges the attainable deviation set and therefore the head's expressivity. The only removed direction is the uniform average $\boldsymbol{v}_{\mathrm{avg}}$, which can be recovered across heads or layers. $\square$

### A.1.3 Proof of Proposition 3.3 (Preservation of Affine Hull and Expressivity).

Let $\{\boldsymbol{v}_i\}_{i=1}^t \subset \mathbb{R}^d$, $\boldsymbol{v}_{\mathrm{avg}} = \frac{1}{t} \sum_{i=1}^t \boldsymbol{v}_i$, and $\Delta_i = \boldsymbol{v}_i - \boldsymbol{v}_{\mathrm{avg}}$.

(i) Full softmax / with zero-order term. A single head with full softmax produces

$$\mathcal{R}_{\mathrm{full}} = \Big\{ \sum_{i=1}^t a_i \boldsymbol{v}_i : a_i \geq 0, \sum_{i=1}^t a_i = 1 \Big\} = \mathrm{Conv}\{\boldsymbol{v}_i\} \subseteq \mathrm{Aff}\{\boldsymbol{v}_i\}.$$

(ii) Zero-sum (without zero-order term). If $\sum_{i=1}^t w_i = 0$, then

$$\sum_{i=1}^t w_i \boldsymbol{v}_i = \sum_{i=1}^t w_i (\boldsymbol{v}_{\mathrm{avg}} + \Delta_i) = \boldsymbol{v}_{\mathrm{avg}} \sum_{i=1}^t w_i + \sum_{i=1}^t w_i \Delta_i = \sum_{i=1}^t w_i \Delta_i,$$

hence

$$\mathcal{R}_{\mathrm{zero\text{-}sum}} = \Big\{ \sum_{i=1}^t w_i \boldsymbol{v}_i : \sum_{i=1}^t w_i = 0 \Big\} = \mathrm{Span}\{\Delta_1, \ldots, \Delta_t\}.$$

Conversely, for any $s = \sum_{i=1}^t u_i \Delta_i \in \mathrm{Span}\{\Delta_i\}$, let $\bar{u} = \frac{1}{t} \sum_{i=1}^t u_i$ and define $w_i = u_i - \bar{u}$. Then $\sum_{i=1}^t w_i = 0$ and $\sum_{i=1}^t w_i \boldsymbol{v}_i = \sum_{i=1}^t u_i \boldsymbol{v}_i - \bar{u} \sum_{i=1}^t \boldsymbol{v}_i = \sum_{i=1}^t u_i (\boldsymbol{v}_i - \boldsymbol{v}_{\mathrm{avg}}) = \sum_{i=1}^t u_i \Delta_i = s$. Hence $\mathrm{Span}\{\Delta_i\} \subseteq \mathcal{R}_{\mathrm{zero\text{-}sum}}$, and thus $\mathcal{R}_{\mathrm{zero\text{-}sum}} = \mathrm{Span}\{\Delta_i\}$.

(iii) Stacking and Minkowski sum. For any $\boldsymbol{y} \in \mathrm{Aff}\{\boldsymbol{v}_i\}$ we can write

$$\boldsymbol{y} = \boldsymbol{v}_{\mathrm{avg}} + (\boldsymbol{y} - \boldsymbol{v}_{\mathrm{avg}}),$$

where $\boldsymbol{v}_{\mathrm{avg}} \in \mathrm{Conv}\{\boldsymbol{v}_i\}$ and, since $\sum_i \Delta_i = 0$, we have $\boldsymbol{y} - \boldsymbol{v}_{\mathrm{avg}} \in \mathrm{Span}\{\Delta_i\}$. Therefore

$$\mathrm{Conv}\{\boldsymbol{v}_i\} + \mathrm{Span}\{\Delta_i\} = \mathrm{Aff}\{\boldsymbol{v}_i\}.$$

Combining (i)–(iii) gives the claimed reachable sets for single heads and their equality to the affine hull when stacked. $\square$

### A.1.4 Proof of Lemma 3.4 (Numerical Stability of Zero-Sum Softmax).

Assume $\|\boldsymbol{v}_i\| \leq B$ for all $i$. By the triangle inequality,

$$\Big\| \sum_{i=1}^t w_{t,i} \boldsymbol{v}_i \Big\| \leq \sum_{i=1}^t |w_{t,i}| \, \|\boldsymbol{v}_i\| \leq B \, t \, \max_i |w_{t,i}|.$$

By the zero-sum softmax construction (see the main text), under bounded logits we have

$$|w_{t,i}| \leq \max\Big\{ \tfrac{1}{t}, \tfrac{|\delta_{t,i}|}{t}, \tfrac{|\rho_{t,i}|}{2t} \Big\}, \qquad \delta_{t,i} = s_{t,i} - \tfrac{1}{t} \sum_{j=1}^t s_{t,j},$$

and $\delta_{t,i}, \rho_{t,i} = O(1)$. Hence $\max_i |w_{t,i}| = O(1/t)$, and therefore

$$\Big\| \sum_{i=1}^t w_{t,i} \boldsymbol{v}_i \Big\| \leq B \, t \cdot O\big(\tfrac{1}{t}\big) = O(B),$$

which is independent of $t$. $\square$

### A.1.5 Proof of Proposition 3.5 (Uniform Lipschitz Bound with $1/\sqrt{t}$ Decay).

Let $\boldsymbol{o}_t(\boldsymbol{x}) = t^{-1/2} \sum_{i=1}^{t} w_{t,i}(\boldsymbol{x}) \, \boldsymbol{v}_i$ with $\sum_{i=1}^{t} w_{t,i}(\boldsymbol{x}) = 0$ and $\|\boldsymbol{v}_i\| \leq B$. For any $\boldsymbol{x}, \boldsymbol{x}'$,

$$\|\boldsymbol{o}_t(\boldsymbol{x}) - \boldsymbol{o}_t(\boldsymbol{x}')\| = \frac{1}{\sqrt{t}} \Big\| \sum_{i=1}^{t} \big(w_{t,i}(\boldsymbol{x}) - w_{t,i}(\boldsymbol{x}')\big) \boldsymbol{v}_i \Big\|$$

$$\leq \frac{1}{\sqrt{t}} \sum_{i=1}^{t} |w_{t,i}(\boldsymbol{x}) - w_{t,i}(\boldsymbol{x}')| \, \|\boldsymbol{v}_i\| \leq \frac{B}{\sqrt{t}} \sum_{i=1}^{t} |w_{t,i}(\boldsymbol{x}) - w_{t,i}(\boldsymbol{x}')|.$$

By the Lipschitz assumption on the weights, $|w_{t,i}(\boldsymbol{x}) - w_{t,i}(\boldsymbol{x}')| \leq (L_w/t) \, \|\boldsymbol{x} - \boldsymbol{x}'\|$, hence

$$\|\boldsymbol{o}_t(\boldsymbol{x}) - \boldsymbol{o}_t(\boldsymbol{x}')\| \leq \frac{B}{\sqrt{t}} \sum_{i=1}^{t} \frac{L_w}{t} \, \|\boldsymbol{x} - \boldsymbol{x}'\| = \frac{B \, L_w}{\sqrt{t}} \, \|\boldsymbol{x} - \boldsymbol{x}'\|.$$

Thus the head is uniformly Lipschitz with constant $BL_w/\sqrt{t}$. $\qquad\qquad\square$

### A.1.6 Implementation of ZeroS Softmax Attention (ZeroS-SM)

**Zero-sum for Standard Softmax Attention (ZeroS-SM)** As shown in Figure 5, our reweighted zero-sum approach can be directly applied to standard softmax attention using logits $s_{t,i} = \boldsymbol{q}_t \boldsymbol{k}_i^\top / \sqrt{d}$, with matrix form:

$$\mathbf{S} = \frac{1}{\sqrt{d}} \mathbf{Q} \mathbf{K}^\top + \mathbf{M}, \ \mathbf{A} = \mathrm{softmax}(\mathbf{S}), \ \mathbf{u} = \big[1, \tfrac{1}{2}, \ldots, \tfrac{1}{N}\big]^\top, \ \bar{\mathbf{S}} = \mathrm{diag}(\mathbf{u}) \, (\mathbf{S} \, \mathbf{1}_N)$$

$$\boldsymbol{\Delta} = \mathrm{diag}(\mathbf{u}) \, \big(\mathbf{S} - \bar{\mathbf{S}} \, \mathbf{1}_N^\top\big), \ \boldsymbol{\varepsilon} = \mathbf{A} \ - \ \mathrm{diag}(\mathbf{u}) \, \mathbf{1}_N \, \mathbf{1}_N^\top \ - \ \boldsymbol{\Delta}$$

$$\mathbf{W} = (\mathbf{g}^1 \, \mathbf{1}_N^\top) \odot \boldsymbol{\Delta} \ + \ (\mathbf{g}^h \, \mathbf{1}_N^\top) \odot \boldsymbol{\varepsilon}, \ \mathbf{O} = \mathbf{W} \, \mathbf{V}$$

## A.2 Runtime Efficiency of the ZeroS Implementation

In recurrent (scan) form, ZeroS maintains three $d \times d$ hidden-state bases

$$e^{s_i} \, \hat{\mathbf{k}}_i^\top \mathbf{v}_i, \quad s_i \, \hat{\mathbf{k}}_i^\top \mathbf{v}_i, \quad \hat{\mathbf{k}}_i^\top \mathbf{v}_i,$$

and reads them out with query-dependent gates; the outputs are summed. While a single fused CUDA kernel is not implemented, we obtain a practical implementation by invoking an existing linear-attention scan three times with different key/value bases:

```
# prepare reweighted queries q1, q2, q3 from q ...
out1 = run_linattn(q1, k * s_i_exp, v, mode='fused_chunk')  # e^{s_i}
out2 = run_linattn(q2, k * s_i,     v, mode='fused_chunk')  # s_i
out3 = run_linattn(q3, k,           v, mode='fused_chunk')  # 1
out  = out1 + out2 + out3
```

We replace the attention layer in a GPT-2 style Transformer (hidden size = 768, 12 heads, 12 layers) with various alternatives and evaluate at sequence length 1024. Baselines include implementations from the same library: LinAttn, GatedLinAttn, HGRN2, RWKV6, RWKV7, and softmax attention (naïve and FlashAttention). All runs use a single NVIDIA L40S, batch size 8, FP32. We report mean latency after warm-up. "Fwd" denotes full-sequence inference (no KV/hidden-state cache). As shown in Table 7. Under this three-scan implementation, ZeroS attains latency, throughput, and memory usage within the range of established linear-attention variants and close to FlashAttention on this setup.

## A.3 Additional Baselines and Ablations

**Setup.** We augment the main results with recent sequence-modeling baselines: Mamba2, Hawk, GatedDeltaNet, and HedgeDog. The evaluation protocol, datasets, and metrics follow the main text.

| Model | FwdLat (s) | FwdStd | TrainLat (s) | TrainStd | Thr.Fwd (tok/s) | Thr.Train (tok/s) | MemFwd (GB) | MemTrain (GB) |
|---|---|---|---|---|---|---|---|---|
| Softmax Attn (naïve) | 0.1306 | 0.0006 | 0.3334 | 0.0009 | 62,740.89 | 24,574.18 | 9.61 | 10.34 |
| RWKV7 | 0.0876 | 0.0016 | 0.2626 | 0.0010 | 93,491.90 | 31,199.35 | 9.81 | 10.61 |
| RWKV6 | 0.0761 | 0.0005 | 0.2252 | 0.0010 | 107,653.00 | 36,382.92 | 9.49 | 9.62 |
| ZeroS | 0.0720 | 0.0008 | 0.1974 | 0.0011 | 113,855.38 | 41,491.29 | 7.48 | 7.61 |
| HGRN2 | 0.0672 | 0.0013 | 0.1480 | 0.0009 | 121,955.16 | 55,336.55 | 6.14 | 6.43 |
| LinAttn | 0.0666 | 0.0010 | 0.1477 | 0.0009 | 122,949.71 | 55,447.86 | 5.79 | 5.90 |
| Softmax Attn (FlashAttn) | 0.0651 | 0.0014 | 0.1473 | 0.0008 | 125,836.28 | 55,620.75 | 5.45 | 5.56 |
| GatedLinAttn | 0.0600 | 0.0009 | 0.1331 | 0.0008 | 136,633.08 | 61,533.68 | 5.74 | 5.89 |

Table 7: Latency, throughput, and peak GPU memory on GPT-2 (768/12/12), sequence length 1024, batch size 8, FP32 on a single L40S. "Fwd" = full-sequence inference without caches.

| Model | Compress | FuzzyRecall | In-ContextRecall | Memorize | NoisyRecall | SelectiveCopy | Average |
|---|---|---|---|---|---|---|---|
| ZeroS (Lin) | 44.0 | 14.9 | 99.9 | 88.1 | 96.1 | 97.8 | 73.5 |
| LinAttn | 33.1 | 8.2 | 91.0 | 74.9 | 75.6 | 93.1 | 62.3 |
| ZeroS (SoftmaxAttn) | 45.2 | 28.0 | 100.0 | 84.3 | 96.6 | 98.5 | 75.4 |
| SoftmaxAttn | 51.6 | 29.8 | 94.1 | 85.2 | 86.8 | 99.6 | 74.5 |
| Mamba2 | 43.6 | 21.1 | 96.4 | 86.9 | 96.7 | 93.3 | 73.0 |
| Hawk | 47.7 | 13.6 | 93.0 | 91.3 | 93.0 | 77.0 | 64.5 |
| GatedDeltaNet | 45.0 | 29.8 | 100.0 | 80.2 | 100.0 | 94.3 | 74.9 |
| HedgeDog | 43.2 | 17.9 | 55.9 | 83.4 | 46.0 | 98.4 | 57.4 |
| *Ablation Study* | | | | | | | |
| ZeroS | 44.0 | 14.9 | 99.9 | 88.1 | 96.1 | 97.8 | 73.5 |
| w/o Angular | 39.5 | 8.5 | 42.8 | 54.5 | 44.8 | 63.3 | 42.2 |
| Ang: w/o PosEmb | 35.8 | 9.4 | 73.3 | 46.2 | 66.2 | 45.8 | 46.1 |
| Ang: additive PosEmb | 38.1 | 14.2 | 94.1 | 86.6 | 87.2 | 93.8 | 69.0 |
| w/o Radial | 35.9 | 9.6 | 84.8 | 86.3 | 86.5 | 92.3 | 65.9 |
| Rad: $u_i$ (linear proj) | 41.2 | 15.5 | 91.9 | 88.3 | 86.1 | 97.2 | 70.0 |
| Rad: $u_i$ (quad form) | 40.9 | 15.4 | 92.6 | 86.6 | 90.8 | 98.5 | 70.8 |
| Rad: $u_i$ (2-distance) | 40.1 | 14.8 | 97.6 | 82.5 | 93.6 | 97.8 | 71.1 |
| Rad: $u_i$ (averaging) | 41.0 | 15.0 | 99.9 | 93.3 | 89.6 | 98.5 | 73.0 |

Table 8: Additional baselines and ablations on six MAD tasks; higher is better.

**Results.** Table 8 reports task accuracies (%). ZeroS attains high scores on In-Context Recall and Noisy Recall and yields a strong overall average. Ablations indicate that removing the angular component substantially degrades performance; additive positional embeddings help but do not match RoPE; and the default radial scoring (with negative similarity) achieves the best average among radial variants.

### A.4 Illustrative Zero-Sum Construction Examples

**Setup.** Let $\{\mathbf{v}_i\}_{i=1}^t \subset \mathbb{R}^d$. A single softmax-attention layer produces $\mathbf{o} = \sum_i \alpha_i \mathbf{v}_i$ with $\alpha_i \geq 0$ and $\sum_i \alpha_i = 1$, hence $\mathbf{o} \in \mathrm{Conv}(\{\mathbf{v}_i\})$. A single ZeroS layer can produce signed, zero-sum combinations $\mathbf{o} = \sum_i w_i \mathbf{v}_i$ with $\sum_i w_i = 0$. Below we list simple sequence-to-sequence mappings that are not representable by a single softmax-attention layer but are representable by a single ZeroS layer.

**Example: Two-token difference.** Target $\mathbf{o} = \mathbf{v}_1 - \mathbf{v}_2$. Softmax requires $\alpha_1 = 1, \alpha_2 = -1$ (invalid). ZeroS: $w_1 = 1$, $w_2 = -1$, others 0.

**Example: Difference from the mean.** Target $\mathbf{o} = \mathbf{v}_1 - \frac{1}{t}\sum_{i=1}^t \mathbf{v}_i$. Softmax needs negative mass on $\{\mathbf{v}_i\}_{i>1}$ (invalid). ZeroS: $w_1 = 1 - \frac{1}{t}$ and $w_{i>1} = -\frac{1}{t}$ (so $\sum_i w_i = 0$).

**Example: Alternating differences.** For even $t$, target $\mathbf{o} = \sum_{i=1}^{t/2}(\mathbf{v}_{2i-1} - \mathbf{v}_{2i})$. Softmax cannot realize alternating $\pm$ weights in one layer. ZeroS: $w_{2i-1} = 1$, $w_{2i} = -1$ (others 0).

### A.5 Additional Dateset Description

#### A.5.1 MQAR

We adopt the Multi-Query Associative Recall (MQAR) task introduced by [56] to characterize a model's performance on repeated, input-dependent lookups over a large vocabulary in a single

forward pass. In the classic associative recall (AR) problem, we store a small, static dictionary of key-value pairs and issue a single, fixed-position query. MQAR generalizes AR by interleaving multiple key-value pairs, each encoded as two consecutive tokens $(k_j, v_j)$, and allows multiple query tokens anywhere in a sequence of length $N$. Formally, given

$$\mathbf{x} = (x_0, ..., x_{N-1}), \qquad x_i \in \mathcal{V}$$

whenever $x_i = k_j$ for some $j < i$, the correct output is

$$y_i = v_j = x_{j+1}$$

and the model must satisfy this for all $1 \le i < N$. By requiring repeated lookups at arbitrary positions, MQAR provides a sharp test of dynamic routing and associative recall, directly contrasting these mechanisms with softmax attention's flexibility and capacity to handle multiple simultaneous queries.

### A.5.2 REGBENCH

RegBench [57] is a synthetic in-context learning benchmark that evaluates a model's ability to infer the structure of regular languages from only a few example strings provided in the prompt. Each problem instance presents $K \in [10, 20]$ example strings $\{d_1^{(i)}, \ldots, d_K^{(i)}\}$ drawn from the same stochastic regular language $L^{(i)}$ defined by a probabilistic finite automaton (PFA). To construct the PFA, RegBench samples a minimal deterministic finite automaton (DFA), the canonical formalization of regular languages. RegBench draws

$$n \sim \text{Uniform}(4, 12), \quad c \sim \text{Uniform}(4, 18), \quad m_i \sim \text{Uniform}(1, 4),$$

and then samples a language-specific alphabet $\Sigma$ of size $c$ uniformly without replacement from a global symbol set of size $c_{\max} = 18$. Define the state set $\mathcal{S} = \{S_1, \ldots, S_n\} \cup \{S_0\}$ with accepting subset $\mathcal{S}_a = \{S_1, \ldots, S_n\}$. For each $S_i$, uniformly without replacement select $m_i$ symbols $x_j \in \Sigma$ and $m_i$ target states $S_j \in \mathcal{S} \setminus \{S_i\}$ to form edges $(S_i, x_j, S_j)$, send all other symbols to $S_0$, and minimize via Hopcroft's algorithm to obtain the canonical DFA $A'$. The PFA inherits $A'$'s topology, assigning

$$T(S_i, x_j, S_j) = \frac{1}{m_i}, \quad T(S_i, x', S') = 0 \quad \text{otherwise,}$$

so that $\sum_{a \in \Sigma} \sum_{s' \in \mathcal{S}} T(s, a, s') = 1, \quad \forall s \in \mathcal{S}.$. From this PFA, $K$ strings of length $\ell \sim$ Uniform$(1, 50)$ are sampled from $S_0$ by simulating $(x_t, S_t) \sim T(S_{t-1}, \cdot, \cdot)$ and concatenating $x_1 x_2 \cdots x_\ell$. Models then perform greedy next-token predictions

$$\hat{x}_j = \arg\max_x p_\theta\big(x \mid \mathbf{d}_{<j}^{(i)}\big),$$

and we report *DFA accuracy* as in Akyürek et al. (2024)

$$\text{accuracy}(p_\theta, L_i) = \frac{1}{\text{NT}} \sum_{\mathbf{d}^{(i)}} \sum_j [\mathbf{1}\big[\hat{x}_j \in \{x' : L_i(x' \mid \mathbf{d}_{<j}^{(i)}) > 0\}\big]],$$

where NT is the total number of tokens in the test set and $L_i(x' \mid \mathbf{d}_{<j}^{(i)})$ is the probability of predicting $x'$ following context $\mathbf{d}_{<j}^{(i)}$ in the language $L_i$. We consider DFA accuracy, the fraction of predictions that correspond to valid transitions in the original DFA, as a direct measure of how faithfully the model has internalized the underlying regular-language structure.

### A.5.3 MAD

We evaluate our proposed architecture using the Mechanistic Architecture Design (MAD) framework, a recently developed methodology for cost-effective evaluation of deep learning architectures [54]. MAD consists of a suite of capability-targeted benchmarks, including in-context recall, fuzzy recall, selective copying, and compression, that probe fundamental sequence modeling capabilities. This approach has been rigorously validated through extensive experimentation spanning over 500 language models from 70M to 7B parameters, demonstrating a strong correlation between performance on these targeted synthetic tasks and compute-optimal perplexity at scale. Through the employment of MAD, which serves as a reliable predictor of large-scale performance, we identify performance advantages without the need for prohibitive computational resources typically associated with architecture validation.

### A.5.4 WikiText-103

WikiText-103 [68] is a large-scale language modeling dataset of over 103 million words compiled from 23,805 *Good* and 4,790 *Featured* Wikipedia articles that have been reviewed by humans, represent broad coverage, and meet common editorial standards. The dataset has long context windows, a large vocabulary of 267,735 types, and requires preservation of case, punctuation, and numerical information so that WikiText-103 accurately reflects the challenges of real-world text.

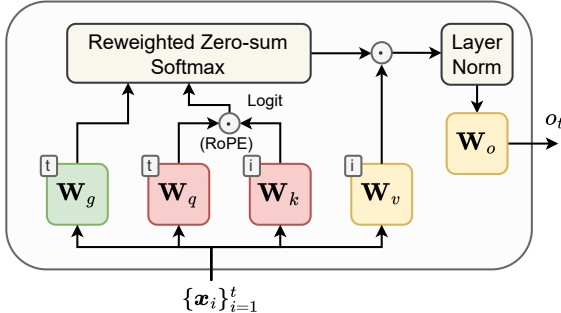

Figure 5: Block Architecture of The ZeroS-SM Layer

### A.5.5 OpenWebText2

OpenWebText2[1] is a large-scale, cleaned and deduplicated web-text corpus created as an open reproduction of OpenAI's WebText dataset: URLs are first extracted from all Reddit submissions with a combined score greater than 3, then scraped, filtered, and deduplicated at both URL and document levels (using MinHash-LSH) to remove low-quality or redundant content. The resulting "plug-and-play" release comprises 17,103,059 documents (around 65.86 GB uncompressed), covering Reddit submissions from 2005 through April 2020, and serves as a high-diversity, up-to-date pretraining corpus for large language models. We use the code environment provided by nanoGPT[2] to implement this dataset training.

### A.5.6 Time Series

We evaluate our module on the time series forecasting benchmark datasets below, following the experimental setup of [65]. **(1) Weather** [69][3]: 21 meteorological variables (e.g., temperature, humidity) collected every 10 minutes in 2020 from a weather station in Germany. **(2) Solar** [70][4]: Solar power outputs recorded every 10 minutes in 2006 from 137 photovoltaic plants in the U.S. **(3) ETT** [71][5]: Transformer load and temperature data sampled at 15-minute (ETTm1/ETTm2) and hourly (ETTh1/ETTh2) intervals from July 2016 to July 2018, including 7 key operational features.

### A.6 Additional Description of Experimental Settings

Our detailed experimental setup is available at the provided code repository. All experiments introduced in this paper can be run on a single Nvidia RTX 4090. For faster training, we parallelize experiments across multiple GPUs. In all benchmarks, we replace the multi-head attention layers with ZeroS layers without modifying any other settings. We do not apply first-layer 0-th order term correction or the $1/\sqrt{t}$ variance scaling in any of our experiments.

### A.7 Impact Statement

This paper introduces an efficient attention mechanism for transformer-based models. As a fundamental architectural improvement, ZeroS primarily affects upstream model capabilities rather than specific applications. The positive impacts include potential reductions in computational costs when processing long sequences. Like most foundational ML research, this work could indirectly contribute to both beneficial and potentially harmful applications depending on how downstream models implement it. However, as an architectural component rather than a deployed system, ZeroS itself poses minimal direct societal concerns.

---

[1] https://openwebtext2.readthedocs.io/en/latest
[2] https://github.com/karpathy/nanoGPT
[3] https://www.bgc-jena.mpg.de/wetter/
[4] http://www.nrel.gov/grid/solar-power-data.html
[5] https://github.com/zhouhaoyi/ETDataset

