# OpenReview forum: "ZeroS: Zero‑Sum Linear Attention for Efficient Transformers"
_NeurIPS.cc/2025/Conference — NeurIPS 2025 spotlight_

### Official Review · Reviewer_FheY · 2025-07-02

**Clarity:** 2
**Significance:** 2
**Originality:** 3
**Rating:** 4
**Confidence:** 4

**Summary:**

The paper proposes a new variant of the self-attention function, both for the linear and the softmax variant. The new self-attention variant is based on a zero-sum attention weights, instead of the traditional convex attention weights used in softmax attention and linear attention variants. The authors show that the resulting zero-sum attention weights are more expressive, in the sense that the sub-space spanned by the zero-sum attention also covers the convex subspace achieved by the convex attention. Furthermore, in the linear attention model, the paper proposes separating the radial and angular components of the self-attention, with the radial component computed through a Taylor-expansion of the softmax attention, and the angular components computed through vanilla linear attention with QK-norm.

The performance of the proposed models are investigated on multiple benchmarks, including recall tasks, language modeling, time-series forecasting, and image classification. Furthermore, an ablation study is performed on some of the components of the models.

**Questions:**

- Can the authors provide examples of tasks that can be solved with the proposed zero-sum softmax, but not the convex softmax? This can significantly help with the credibility of the proposed idea or zero-sum attention.
- Can the authors provide further benchmark results, such as the needle in the haystack, arithmetic, and sequence length benchmarking results to further solidify the superiority of the proposed model?
- What would happen if instead of the $u_i$s, you used the query/key in the radial component for calculating $\delta$? Does the model perform worse? If so, by how much? I’m curious because following the logic of the paper, you can see the magnitude of the query and the key vectors as the input of the radial component, and their angle as the input to the angular component.
- The notation in the final paragraphs of the third section gets a little bit confusing. Is $H_t$ a scalar or a matrix? Are you computing the inner products of the key and value in $F_t$, $G_t$, $H_t$? If so, why change it compared to the linear attention model?
- Can you perform some ablation studies explaining how much of the performance contribution comes from the angular component and how much of the performance contribution from the radial component?
- Can you provide the proofs for the theorems in the main body?
- Also, there is a typo in proposition 3.1.

**Ethical Concerns:**

["NO or VERY MINOR ethics concerns only"]

**Final Justification:**

I think the paper proposes an interesting approach with the zero-sum softmax, which is supported by helpful and intuitive theoretical discussions. During the rebuttal, some of my concerns were addressed. Notably, the authors provided theoretical and empirical evidence supporting the benefits of using the zero-sum softmax attention, an ablation on the components, further clarification on their motivation behind the gating mechanism, and full proof for the theorems. They've also noted that they intend to improve the notations used in the paper in the final version. This will admittedly be a relatively large overhaul of the paper.

The only concern of mine which was not addressed during the rebuttal is the main intent of the paper. Specifically, I do not understand whether the paper should be taken as a new variant of the softmax attention, or a linear attention model. If the latter is the aim, then some of the results in the paper (such as the normalization theorem) seem to be more window-dressing than helpful and informative results. If the intent is the former, the authors should have leaned more towards comparing themselves with recent variants of softmax attention, which seem to be completely missing from the results. Therefore, I lean towards acceptance, but I do not have a strong opinion about the paper.

**Limitations:**

I think there are two limitations concerning this paper. Firstly, the motivation behind the main contributions of the paper is not well-established. I am not convinced that the zero-sum attention mechanism is necessary, despite the fact that it provably provides a more expressive self-attention model. Secondly, the paper suffers from a lack of an organized benchmarking effort. I believe the authors should focus on the most recent linear attention variants, and compare their proposed method with those models on all tasks.

**Paper Formatting Concerns:**

No formatting issues.

**Quality:**

2

**Strengths And Weaknesses:**

**Strengths.** The paper starts with a convincing argument about the shortcomings of the linear attention model, citing the relevant studies trying to alleviate these issues. The notations used in the earlier sections of the paper are clear and helpful, and the paper provides a helpful summary of the investigated architecture. The related works section is well integrated into the main contributions of the paper. The theoretical claims seem to be correct, although the proofs for the theorems are not provided. The paper provides an illustration of the proposed method, thus helping the reader to understand the new components in places where the notations fail to do so.

**Weaknesses.** I believe the main issue of the paper is a lack of well-established motivation for the architectural innovations that are at the center of the proposed models. Specifically, the following claims are not well-supported:
- The need for a more expressive attention mechanism is not supported by the paper properly. It is true that the zero-sum attention is more expressive than a convex attention. However, one could also argue that the original idea behind the attention mechanism tries to establish an inductive bias by this choice. The authors may point to the better performance of the model in certain benchmarks (although, the experiment results are more of a mix-bag), but in some of the experiments the model is actually either in the under-parameterized regime or near it. Consequently, having a more complex model usually results in better performance.
- The importance of having the negative angular signal from the self-attention mechanism is also not supported beyond the “more expressive” argument. Do the authors have experimental results showing that this is a particularly important issue? For instance, can you perform an experiment wherein you limit the angle between the query and the key in a softmax transformer to the positive range and show significant drop in performance?
- The gating on the softmax components seem to be coming out of nowhere in the paper, with very limited setup or justification. What is your reason behind this architecture decision? How can you support its importance?
- The contribution corresponding to the normalization factor is undermined by the fact that there’s extensive research on this component in the linear attention literature. In Mamba2 and the DeltaNet family of models, the post-norm component seems to be eliminating the need for any explicit normalization. In fact, you seem to be using this component in your own architecture.
- The decision behind introducing the $u_i$ factors in the radial component is not clearly established. Why is it necessary for the $\delta$ elements to provide information about the “deviation of step $i$ relative to the previous steps?”

The paper also seems to be suffering from a mixed message problem. Initially, the paper appears to be focusing on providing an answer for the long-standing question in the community about shortcomings of the linear attention model. However, the focus shifts multiple times from the softmax function itself (and how to improve it) to the linear attention and back. And ultimately, in the experiments there is a clear shift towards an emphasis on the softmax variant with the zero-sum structure. While interesting as a secondary investigation for the appendix, I see this decision as a distraction, occupying a space in the main body that could be filled with better experiments on the linear model.

Furthermore, the experiment results are generally a mixed-bag. The model doesn’t seem to be providing significant improvement compared to the baselines, with marginal improvements in certain benchmarks. For instance, in the MAD benchmark the model rarely beats the linear attention variants by a significant margin, and considering the outdated list of baselines, it doesn’t paint a promising picture. Furthermore, there is inconsistency in the list of baselines used in the experiments, with some baselines missing from certain experiments without explanation. I think it would be very helpful if the authors could provide the performance of a consistent set of baselines from the most recent variants of linear attention. For instance, use Gated DeltaNet instead of DeltaNet, and Mamba2 instead of Mamba1, and so on. Furthermore, there are certain models like hedgehog that also aim to find the “secret sauce” of the softmax attention instead of going the "approximation" way in order to improve linear attention. I believe you should also include these models in your list of baselines.

Moreover, considering the number of extra components introduced in the proposed architecture, I expected a much more thorough ablation study. For example, it would be helpful to have an experiment showing the effect of the angular component. The angular component seems to be vanilla linear attention coupled with QK norm and RoPE.

---

> ### Author Rebuttal · Authors · 2025-07-31
>
> ```
> ... motivation for the innovations ...
> ```
>
> Thank you very much for your valuable suggestions, which help improve the paper's clarity and accessibility. We will expand on the following points:
>
> ```
> more expressive ... inductive bias by this choice ... in the under-parameterized regime
> ```
> Yes, softmax ensures convexity via a zero-order term ($1/t$), which stabilizes outputs by compressing them into [0,1]. This introduces a bias toward averaging, suitable for simpler tasks like *Fuzzy Recall (FR) and Compression* in MAD. In contrast, complex tasks such as *In-Context Recall (ICR) and Noisy Recall (NR)* require precise retrieval and benefit from higher-order interactions, where ZeroS shows clear gains (see Table 1), aligning with our goal of enhancing expressivity by removing this constant bias.
>
> Including or removing the $1/t$ term reflects a trade-off in parameter efficiency: keeping it allows one-layer averaging but requires multiple layers for differencing, while removing it enables one-layer differencing but multiple layer for averaging (see Proposition 3.3). The latter is often preferable, as averaging is simpler to approximate (parameter-free average-pooling is enough), while differencing is harder to learn.
>
> As Table 1 shows, ZeroS improves significantly on *ICR and NR*, while gains on *FR and Compression* are smaller: supporting our view that this reflects a true shift in modeling capability preferences, not enough to be explained only by under-parameterization.
>
> ```
> limit the angle between the query and the key
> ```
>
> Our early experiments and prior works like RetNet and GLA suggest that lifting the non-negativity constraint on queries and keys (i.e., relaxing convexity) improves performance (one of the key motivations behind our design).
>
> As a test, we applied a softplus function to enforce positive queries and keys, limiting the sign of $\cos \theta$ as follows. This led to a clear drop in performance, especially on complex tasks like *ICR and NR*, showing the importance of signed angular interactions for expressivity.
>
>
> |Model|Compress|FuzzyRecall|In-ContextRecall|Memorize|NoisyRecall|SelectiveCopy|Average|
> |---|---|---|---|---|---|---|---|
> |ZeroS|44.0|14.9|99.9|88.1|96.1|97.8|73.5|
> |ZeroS(QK ForcePositive)|41.3|14.5|79.3|65.4|72.3|82.2|59.2|
> |SoftmaxAttn|51.6|29.8|94.1|85.2|86.8|99.6|74.5|
> |SoftmaxAttn(QK ForcePositive)|51.6|29.5|80.3|80.5|74.5|99.1|69.3|
>
> ```
> gating coming out of nowhere in the paper
> ```
>
> We apologize for any confusion caused by our modular presentation, intended to help readers navigate key components. Below is a concise, chronological summary of the motivation. Brackets [ ] indicate our design insights; parentheses ( ) refer to prior findings:
>
> ---
>
> [Consider the Taylor expansion of softmax]
>
> ↓
>
> (Prior works approximate softmax using low-order terms)
> +
> (Others note that convex weights (due to non-negativity) limit expressivity but improve stability)
>
> ↓
>
> [The zero-order term ($1/t$) enforces convexity; higher-order terms yield zero-sum weights for richer interactions]
>
> ↓
>
> [Removing the zero-order term allows negative weights (more expressivity) while retaining stability via normalized zero-sum structure]
> +
> (Differential Transformer [3] and others may benefit implicitly from this)
>
> ↓
>
> [Our experiments confirm that zero-sum softmax (without $1/t$) improves performance]
>
> ↓
>
> [**Removing the zero-order term = applying a fixed gating of 0**, suggesting learnable gating on higher-order terms for more expressivity]
>
> ↓
>
> [Experiments support this, leading to the final gated ZeroS design]
>
> ---
>
> ```
> the normalization...
> ```
>
> Yes, that's correct. Lemma 3.4 and Proposition 3.5 provide a theoretical guarantee of numerical stability for zero-sum attention, showing that a simple $1/\sqrt{t}$ normalization ensures stability across sequence lengths.
>
> While many linear attention models use output LayerNorm to stabilize training—making $1/\sqrt{t}$ optional—LayerNorm is an **engineering fix**, whereas our results offer a **theoretical foundation**.
>
> For instance, standard linear attention may explode under $-\infty$ weights, while ZeroS weights (e.g., from $s_i$ without gating), which only subtract $1/t$ from softmax, remain stable. Although ZeroS uses prefix scans on  KV that can't actually handle $-\infty$ due to this **engineering constraints**, this does not weaken its **stronger theoretical robustness** over standard linear attention.
>
> ```
> Recent baselines...; Ablation of angular part...; Why $u_i$ factors...
> ```
>
> We summarize the requested results in the table below.
>
> We added recent baselines including [Mamba2, Hawk, Gated DeltaNet, xLSTM, HedgeDog], using implementations from the authors of [1] and our codes for HedgeDog for a linear approximation of softmax. ZeroS performs strongly on complex tasks like *ICR and NR*, and outperforms models like RWKV and xLSTM that heavily compress or alter hidden states on *Memorize*, leading to a higher average score.
>
> HedgeDog’s results are particularly interesting: it performs well on soft-averaging tasks (*Compress, FR, Memorize*), but drops sharply on *ICR and NR*, likely because it captures mainly the $1/t$ term from softmax while failing to approximate higher-order terms: supporting our core paper motivation.
>
> The angular part in ZeroS follows standard linear attention. Removing it (w/o Angular) eliminates query-key bilinear interactions, reducing hidden state from $3 \times d \times d$ to $3 \times d$, and thus significantly reducing expressiveness. Reintroducing QK but removing positional embeddings gives only marginal gains, as $\cos \theta_{t,i}$ lacks positional cues. Additive PosEmb improve performance but still fall short of RoPE.
>
> As for computing $s_i$ from $u_i$ in the radial part: since zero-sum weights model higher-order variation, $s_i$ should reflect such structure. It can be derived locally (e.g., linear projection, quadratic form), or from context (e.g., window mean, deviation, or negative similarity with mean). Since ZeroS targets outlier interactions, negative similarity is especially effective.
>
> Our code supports all variants, and experiments confirm that negative similarity yields the best results (see Rad: u_i), as briefly noted in the main text.
>
> |Model|Compress|FuzzyRecall|In-ContextRecall|Memorize|NoisyRecall|SelectiveCopy|Average|
> |---|---|---|---|---|---|---|---|
> |ZeroS(Lin)|44.0|14.9|99.9|88.1|96.1|97.8|73.5|
> |LinAttn|33.1|8.2|91.0|74.9|75.6|93.1|62.3|
> |ZeroS(SoftmaxAttn)|45.2|28.0|100.0|84.3|96.6|98.5|75.4|
> |SoftmaxAttn|51.6|29.8|94.1|85.2|86.8|99.6|74.5|
> |Mamba2|43.6|21.1|96.4|86.9|96.7|93.3|73.0|
> |Hawk|47.7|13.6|93.0|91.3|93.0|77.0|64.5|
> |GatedDeltaNet|45.0|32.7|100.0|81.7|100.0|95.7|71.0|
> |xLSTM|43.4|47.6|100.0|79.8|100.0|95.4|73.2|
> |HedgeDog|43.2|17.9|55.9|83.4|46.0|98.4|57.4|
> |[AblationStudy↓]|-|-|-|-|-|-|-|
> |ZeroS|44.0|14.9|99.9|88.1|96.1|97.8|73.5|
> |w/o Angular|39.5|8.5|42.8|54.5|44.8|63.3|42.2|
> |Ang: w/o PosEmb|35.8|9.4|73.3|46.2|66.2|45.8|46.1|
> |Ang: w/ additive PosEmb|38.1|14.2|94.1|86.6|87.2|93.8|69.0|
> |w/oRadial|35.9|9.6|84.8|86.3|86.5|92.3|65.9|
> |Rad: u_i - linear proj|41.2|15.5|91.9|88.3|86.1|97.2|70.0|
> |Rad: u_i - quad form|40.9|15.4|92.6|86.6|90.8|98.5|70.8|
> |Rad: u_i - 2-distance|40.1|14.8|97.6|82.5|93.6|97.8|71.1|
> |Rad: u_i - averaging|41.0|15.0|99.9|93.3|89.6|98.5|73.0|
>
>
> Regarding the suggestion on additional benchmarks: due to space and time constraints, and given our experimental coverage, we refer you to relevant existing results. Needle-in-a-haystack behaviors are covered in MAD’s *Selective Copy and ICR*. Arithmetic reasoning is addressed in RegBench and MQAR, both of which extend arithmetic tasks. Sequence length is shown over sub-tasks averaging in synthetic datasets like MQAR.
>
> ```
> tasks that can be solved with the proposed zero-sum softmax, but not the convex softmax...
> ```
>
> Please refer to our responses to Reviewer D2cQ.
>
> ```
> what if we use query/key calculation in the radial component...
> ```
>
> The radial component is designed as a product of two scalars (from steps $t$ and $i$) to enable higher-order zero-sum softmax while keeping $O(N)$ complexity. Using QK dot products instead would require $O(N^2)$, losing linearity. Alternatively, applying channel-wise zero-sum softmax over keys could preserve linearity, but would increase the hidden state to a $3 \times d \times d \times d$ tensor ($O(d^3)$ space). Our follow-up work explores this $O(d^n)$ expressivity-efficiency trade-off, and we’re happy to briefly share this direction here.
>
> ```
> notation in the final paragraphs ... confusing
> ```
>
> This refers to the 3 hidden state matrices. In recurrence, ZeroS performs associative scans over states based on $e^{s_i} \hat{\mathbf{k}}_i^\top\mathbf{v}_i$, $s_i \hat{\mathbf{k}}_i^\top \mathbf{v}_i$, and $\hat{\mathbf{k}}_i^\top \mathbf{v}_i$, then reads them out using different gated queries and sums the results. In short, it's a linear scan over a $3 \times d \times d$ hidden state. Please see our response to Reviewer WRY9 for more.
>
> ```
> could be filled with better experiments on the linear model
> ```
>
> To show the generality of zero-sum mechanisms, we included quadratic models. In the revision, we’ll add more discussion and results on linear models to balance coverage.
>
> ```
> provide the proofs ... a typo in proposition 3.1 ...
> ```
>
> We apologize for omitting full proofs in the submission version due to space constraints. We presented the propositions in a concise form that retains the core reasoning, as the results are relatively straightforward. Full details will be restored in the appendix in the revision. We're also happy to walk through the arguments during discussion. We’ll correct the missing zero-sum condition ($= 0$) in proposition 3.1.
>
> For complexity-related questions, please see our replies to other reviewers.
>
> ---
>
>
> ### References
>
> See the same references listed under our response to Reviewer kvKy.

---

> > ### Comment · Reviewer_FheY · 2025-08-05
> > **Official comment by reviewer FheY**
> >
> > I thank the authors for their comprehensive response.
> >
> > Regarding the gating mechanism, this is an interesting idea. I think it would be helpful if the authors attempted to ease the reader into this idea, by having more emphasis on this connection. Furthermore, I think it would be helpful if the authors could show the value of these gates at convergence (especially for the arithmetic averaging term) so we would better understand where the optimal values may end up.
> >
> > About the normalization component, the connection is clear now. Does it have a similar performance to the layer normalization? The fact that the final normalization component is a sublinear function of t is also very interesting. Does layer normalization also approach the same scale? I suspect it would, since without any gating mechanism, the magnitude of the output can be seen as a function of the sequence size. I think it would be beneficial to the paper if there was a small discussion about this, since right now the normalization section seems a little bit detached from the rest of the paper.
> >
> > Regarding the experimental results, I thank the authors for further discussion. I think considering the limitations of the rebuttal period this year, the results are satisfactory.
> >
> > About an example of a problem only the zero-sum softmax can solve, I think the proposed examples provide further insight into the paper and methodology. Please add them to the paper.
> >
> > Thank you for clarifying the remaining points as well. Please make sure to add the full proofs to the paper, even though the theorems and the mathematical conclusions are more or less straightforward.
> >
> > I think the authors’ response has addressed most of my concerns. However, I think the notation in the paper is still very confusing. I believe the paper would benefit from a major overhaul of the notations and the presentation of the mathematical results. Especially with regards to the sections concerned with implementation details.

---

> > > ### Author Response · Authors · 2025-08-06
> > >
> > > Thank you very much for your kind recognition and valuable suggestions. Your feedback has been essential in improving the clarity and quality of our paper. We will include all points mentioned in the rebuttal in the revision. Below is our response to your remaining detailed comments. Thank you again for the time and effort you've dedicated to reviewing our work!
> > >
> > > ```
> > > ...could show the value of these gates...
> > > ```
> > >
> > > Thank you for the suggestion. In the revision, we will add a section in the appendix that visualizes the learned gating parameters across all attention layers of the trained ZeroS Transformer on NLP tasks. In tasks involving complex high-dimensional feature interactions, the higher-order gating weights will tend to be more strongly activated.
> > >
> > > ```
> > > About the normalization component...
> > > ```
> > >
> > > Here, we provide a more detailed ablation on normalization. The results shown in the table below are consistent with the discussion in our paper:
> > >
> > > 1. The $1/\sqrt{t}$ term offers a theoretical guarantee of numerical stability for ZeroS. Without any normalization, introducing this decay factor helps stabilize training and can improve performance (depending on the sequence length of the task).
> > >
> > > 2. Prior work has shown that applying LayerNorm to the output of linear attention consistently improves performance (regardless of sequence length). Following this, we adopt LayerNorm in our implementation. In this setting, since LayerNorm already ensures numerical stability, adding the $1/\sqrt{t}$ factor does not significantly affect performance. Therefore, in practice, we disable this factor by default and instead present it as a theoretically stronger bound when comparing to other linear attention variants.
> > >
> > > | Model | Compress | Fuzzy Recall | In-Context Recall | Memorize | Noisy Recall | Selective Copy | Average |
> > > |---|---|---|---|---|---|---|---|
> > > | ZeroS (w/ LN) | 44.0 | 14.9 | 99.9 | 88.1 | 96.1 | 97.8 | 73.5 |
> > > | ZeroS (w/o any Normalization) | 39.1 | 12.3 | 89.0 | 87.0 | 91.7 | 97.1 | 69.4 |
> > > | ZeroS (w/ LN and $1/\sqrt{t}$) | 44.2 | 15.2 | 99.3 | 89.3 | 95.8 | 96.3 | 73.4 |
> > > | ZeroS (w/ $1/\sqrt{t}$) | 39.9 | 13.6 | 93.2 | 88.4 | 95.1 | 96.9 | 71.2 |
> > >
> > > ```
> > > the notation ... with regards to the sections concerned with implementation details...
> > > ```
> > >
> > > Thank you for your comment, and we apologize for the confusion. This was likely caused by our attempt to save space by introducing too many **new symbols in Section 3.3**. To improve clarity, we will **revert to more explicit forms** and restate the explanation in expanded form **using original terms introduced before Section 3.3**.
> > >
> > > First, a quick clarification: following the convention in Transformer and linear attention literature, we treat each input step as a **row vector** ($1 \times d$), meaning that $\mathbf{a}^\top \mathbf{b}$ produces a **rank-1 matrix** (outer product), while $\mathbf{a} \mathbf{b}^\top$ gives a **scalar** (dot product). We’ll make this convention explicit in the revised section to avoid cross-domain confusion.
> > >
> > > In standard linear attention, the output at step $t$ is computed as:
> > >
> > > $$
> > > \mathbf{o}\_t = \sum\_{i=1}^t \mathbf{q}\_t \mathbf{k}\_i^\top \mathbf{v}\_i = \mathbf{q}\_t \left[\sum\_{i=1}^t \mathbf{k}\_i^\top \mathbf{v}\_i\right],
> > > $$
> > >
> > > by separating the expression into $t$- and $i$-dependent parts.
> > >
> > > In ZeroS, the attention score $\mathbf{q}_t \mathbf{k}_i^\top$ **is replaced by** a product of the **angular part** $\hat{\mathbf{q}}\_t \hat{\mathbf{k}}\_i^\top$ and a scalar **radial term** $r\_{t,i}$, which takes the form:
> > >
> > > $$
> > > \mathbf{o}\_t = \sum\_{i=1}^t r\_{t,i} \hat{\mathbf{q}}\_t \hat{\mathbf{k}}\_i^\top \mathbf{v}\_i
> > > $$
> > >
> > > $$
> > > r_{t,i} = \frac{\sigma_t^h}{\sum_{j=1}^t \exp(s_j)} \exp(s_i) + \frac{\sigma_t^1 - \sigma_t^h}{t}(s_i - \bar{s}_t) - \frac{\sigma_t^h}{t}.
> > > $$
> > >
> > > To enable associative scan, we **reorganize** the equation to cleanly **separate $t$-dependent and $i$-dependent terms**:
> > >
> > > $$
> > > \mathbf{o}\_t = \sum\_{i=1}^t \left(\left[ \hat{\mathbf{q}}\_t \frac{\sigma\_t^h}{\sum\_{j=1}^t \exp(s\_{j})} \right] \left[\exp(s\_{i}) \hat{\mathbf{k}}\_i^\top {\mathbf{v}}\_i \right] + \left[\hat{\mathbf{q}}\_t \frac{\sigma\_t^1 - \sigma\_t^h}{t}\right] \left[s\_{i} \hat{\mathbf{k}}\_i^\top {\mathbf{v}}\_i \right] + \left[\hat{\mathbf{q}}\_t (\frac{\sigma\_t^h - \sigma\_t^1}{t} \bar s\_t - \frac{\sigma\_t^h}t)\right] \left[\hat{\mathbf{k}}\_i^\top {\mathbf{v}}\_i \right]\right)
> > > $$
> > >
> > > This shows that the expression decomposes into a sum of 3 associative scans over hidden states:
> > >
> > > $$
> > > \left[\exp(s_i)\hat{\mathbf{k}}_i^\top \mathbf{v}_i,\ s_i\hat{\mathbf{k}}_i^\top \mathbf{v}_i,\ \hat{\mathbf{k}}_i^\top \mathbf{v}_i\right],
> > > $$
> > >
> > > which **form a $3 \times d \times d$ hidden state**. Each is read out by different **gated queries**. Since we haven’t implemented a fused CUDA kernel for this $3 \times d \times d$ structure, a practical alternative is to perform three separate runs of standard linear attention, as illustrated in our code:
> > >
> > > (To be continued)

---

> > > > ### Author Response · Authors · 2025-08-06
> > > >
> > > > (Continued from previous)
> > > >
> > > > ```python
> > > > # prepare reweighted query q1, q2, q3 from q...
> > > > out1 = run_linattn(q1, k * s_i_exp, v, mode='fused_chunk')
> > > > out2 = run_linattn(q2, k * s_i, v, mode='fused_chunk')
> > > > out3 = run_linattn(q3, k, v, mode='fused_chunk')
> > > > out = out1 + out2 + out3
> > > > ```
> > > >
> > > > As noted in responses to other reviewers, this approach remains computationally efficient and close in cost to standard linear attention.
> > > >
> > > > We hope this breakdown clarifies how ZeroS is implemented under associative scan, and we will revise Section 3.3 to reduce symbol overload and improve clarity.
> > > >
> > > > ---
> > > >
> > > > ```
> > > > ...add the full proofs to the paper...
> > > > ```
> > > >
> > > > We will include the expanded and reformatted versions of the simplified proofs currently presented in the theorem statements in the appendix of the revision. The separated proofs will be structured as follows.
> > > >
> > > > **Proposition 3.1 (Convex vs. Zero-Sum Span)**
> > > >
> > > > **Proof:**
> > > > We have $\sum_i\alpha_i\mathbf v_i-\mathbf v_{\mathrm{avg}}=\sum_i(\alpha_i-1/t)\mathbf v_i$ with zero-sum weights $w_i=\alpha_i-1/t\ge-1/t$. Thus, clearly $\mathcal C_{\mathrm{dev}}\subseteq\mathcal Z$. For strictness, if vectors are not identical, pick distinct $\mathbf v_j,\mathbf v_k$. Consider $\mathbf v_j-\mathbf v_k\in\mathcal Z$. To represent it in $\mathcal C_{\mathrm{dev}}$, we need $\alpha_k-1/t=-1\Rightarrow\alpha_k=-1+1/t<0$, which is impossible. Thus, strictness holds.
> > > >
> > > > **Corollary 3.2 (Expressive Gain of Zero-Sum Attention)**
> > > > **Proof:**
> > > > Follows immediately from Proposition 3.1: convex set (softmax) is strictly contained in the zero-sum set (ZeroS).
> > > >
> > > > ---
> > > >
> > > > **Proposition 3.3 (Preservation of Affine Hull and Expressivity)**
> > > > **Proof:**
> > > >
> > > > 1. Full softmax: Outputs form convex combinations $\sum_i a_i \mathbf v_i$ with $\sum_i a_i=1$, clearly giving the convex hull: $
> > > > \mathcal{R}_{\text{full}}=\mathrm{Conv}\\{\mathbf v_i\\}\subseteq\mathrm{Aff}\\{\mathbf v_i\\}.
> > > > $
> > > >
> > > > 2. Zero-sum weights: With weights $w_i$, $\sum_i w_i=0$, output is: $
> > > > \sum_i w_i \mathbf v_i=\sum_i w_i(\mathbf v_{\mathrm{avg}}+\Delta_i)=\sum_i w_i\Delta_i.
> > > > $ Thus: $
> > > > \mathcal{R}_{\text{zero-sum}}=\mathrm{Span}\\{\Delta_i\\}.
> > > > $
> > > >
> > > > 3. Stacking (Minkowski sum): Consider any $\mathbf y\in\mathrm{Aff}\\{\mathbf v_i\\}$. We have a unique decomposition: $
> > > > \mathbf y=\mathbf v_{\mathrm{avg}}+(\mathbf y-\mathbf v_{\mathrm{avg}}),
> > > > $ where clearly $\mathbf v_{\mathrm{avg}}\in\mathrm{Conv}\\{\mathbf v_i\\}$, and $\mathbf y-\mathbf v_{\mathrm{avg}}\in\mathrm{Span}\\{\Delta_i\\}$. Thus: $
> > > > \mathrm{Conv}\\{\mathbf v_i\\}+\mathrm{Span}\\{\Delta_i\\}=\mathrm{Aff}\\{\mathbf v_i\\}.
> > > > $ The stacked network precisely covers the affine hull, proving the proposition.
> > > >
> > > > ---
> > > >
> > > > **Lemma 3.4 (Numerical Stability of Zero-Sum Softmax)**
> > > >
> > > > **Proof:**
> > > >
> > > > By the triangle inequality and boundedness of each $\mathbf v_i$, we have:
> > > >
> > > > $$
> > > > \bigl\|\sum_{i=1}^t w_{t,i}\mathbf v_i\bigr\|
> > > > \le \max_i |w_{t,i}|\sum_{i=1}^t \|\mathbf v_i\|
> > > > \le B t \max_i|w_{t,i}|.
> > > > $$
> > > >
> > > > Since $w_{t,i}=\sigma_t^1\frac{\delta_{t,i}}{t}+\sigma_t^h\varepsilon_{t,i}$ with $\sigma_t^1,\sigma_t^h\in[0,1]$ and bounded logits imply $\delta_{t,i},\varepsilon_{t,i}=O(1)$, it follows that: $
> > > > |w_{t,i}|=O\left(\frac{1}{t}\right).
> > > > $ Thus, the expression simplifies to:
> > > >
> > > > $$
> > > > \bigl\|\sum_{i=1}^t w_{t,i}\mathbf v_i\bigr\|\le B t O\left(\frac{1}{t}\right)=O(B),
> > > > $$
> > > >
> > > > showing stability independent of $t$.
> > > >
> > > > **Proposition 3.5 (Uniform Lipschitz Bound with $1/\sqrt{t}$ Decay)**
> > > >
> > > > **Proof:**
> > > >
> > > > By definition of $\mathbf o_t(\mathbf x)$, applying triangle inequality, boundedness of vectors, and Lipschitz assumption on weights:
> > > >
> > > > $$
> > > > \|\mathbf o_t(\mathbf x)-\mathbf o_t(\mathbf x')\|
> > > > \le\frac{1}{\sqrt t}\sum_{i=1}^t |w_{t,i}(\mathbf x)-w_{t,i}(\mathbf x')|\|\mathbf v_i\|
> > > > \le\frac{B}{\sqrt t}\sum_{i=1}^t\frac{L_w}{t}\|\mathbf x-\mathbf x'\|.
> > > > $$
> > > >
> > > > Since $\sum_{i=1}^t L_w/t = L_w$, this simplifies to:
> > > >
> > > > $$
> > > > \|\mathbf o_t(\mathbf x)-\mathbf o_t(\mathbf x')\|
> > > > \le\frac{B L_w}{\sqrt t}\|\mathbf x-\mathbf x'\|,
> > > > $$
> > > >
> > > > which demonstrates uniform Lipschitz stability decaying as $O(1/\sqrt t)$.

---

> > > ### Author Response · Authors · 2025-08-09
> > >
> > > Dear Reviewer FheY,
> > >
> > > As the rebuttal discussion period is drawing to a close, we would like to follow up on your last concern regarding the clarity of the implementation detail notations. In our previous response, we explained that we followed the common practice in the linear attention literature (such as using row-vector notation), and we proposed a revision for the **implementation part** that **uses only the symbols introduced before Section 3.3**, avoiding any new notation to minimize the reading burden. May we confirm if this revision has addressed your concern? If anything remains unclear, we would be glad to use the remaining time to provide a brief clarification. Once again, thank you for your time and effort in reviewing our work.
> > >
> > > Best regards,
> > > The Authors

---

> > > > ### Comment · Reviewer_FheY · 2025-08-09
> > > > **Official comment by reviewer FheY**
> > > >
> > > > I thank the authors for further clarification. I still think there are some issues with the mixed messaging in the paper regarding softmax attention and linear attention, which I noted in my original review. However, the majority of my other concerns have been addressed by the authors. Therefore, I incline towards accepting the paper.

---

> > > > > ### Author Response · Authors · 2025-08-09
> > > > >
> > > > > Thank you very much for your prompt response and thoughtful suggestions. In our revision, we will make sure to maintain a balanced presentation of the softmax attention + ZeroS content in the main paper, ensuring it does not overshadow the main aspects. We sincerely appreciate your valuable feedback and acknowledgment once again.

---

### Official Review · Reviewer_D2cQ · 2025-07-05

**Clarity:** 3
**Significance:** 3
**Originality:** 3
**Rating:** 4
**Confidence:** 3

**Summary:**

This paper introduces Zero-Sum Linear Attention (ZeroS), a novel linear-time attention mechanism that aims to close the performance gap between efficient transformers and standard softmax attention. The authors identify two key limitations in existing attention mechanisms: 1) the restriction to convex combinations, which prevents subtractive or contrastive operations within a single layer, and 2) a uniform weight bias from the constant zero-order term in the softmax Taylor expansion, which dilutes attention in long contexts.

**Questions:**

- Proposition 3.3 and Corollary 3.2 elegantly show that zero-sum attention expands the set of representable functions beyond a simple convex hull. Can you provide a simple, concrete example of a sequence-to-sequence task that standard attention would struggle with but that zero-sum attention can solve easily in a single layer?

- The zero-sum quadratic version of your model, ZeroS-SM, consistently outperforms the standard Transformer baseline, for instance in Table 1. This suggests that even without the constraint of linear time complexity, removing the zero-order term is beneficial. Do you see any downsides or trade-offs to applying this modification to standard softmax attention in models where quadratic complexity is not a bottleneck?

**Ethical Concerns:**

["NO or VERY MINOR ethics concerns only"]

**Limitations:**

The authors provide a good discussion of limitations in the conclusion, noting that their work prioritizes algorithmic expressivity over low-level engineering optimizations (like custom GPU kernels) and that evaluation on very large-scale LLMs was beyond their resource constraints. These are reasonable limitations. One other potential limitation is the complexity of the final algorithm (Algorithm 1), which, while linear, involves more steps and states to track than simpler linear attention mechanisms. This might pose a barrier to adoption or require more engineering effort to optimize for hardware, as the authors allude to.

**Paper Formatting Concerns:**

No.

**Quality:**

3

**Strengths And Weaknesses:**

## Pros
- The paper provides a compelling and novel diagnosis of a fundamental weakness in softmax and many linear attention mechanisms: the performance-degrading effect of the zero-order uniform term in the softmax expansion. Identifying this term as a source of "attention dilution" and linking it to the limitations of convex combinations is a significant conceptual contribution.

-  The proposed solution—to simply remove the zero-order term and reweight the residuals—is both simple and powerful. The authors provide strong theoretical support for this approach, proving that the resulting zero-sum weights increase expressivity (Proposition 3.1) while maintaining numerical stability (Lemma 3.4, Proposition 3.5).

- The ablation studies in Tables 5 and 6 effectively demonstrate the importance of each component of the ZeroS framework. The results clearly show that removing the zero-sum reweighting (w/o RWSM) or the gating mechanism leads to a performance drop, validating their design choices.

## Cons
- While the core idea is simple, the final implementation of ZeroS is quite complex, involving several components: the calculation of deviation logits with a smoothing prior, the separation and gating of first-order and higher-order softmax residuals, and the reintroduction of an angular component. This complexity could make the method harder to analyze and adopt compared to simpler linear attention mechanisms.

- The method decouples the attention weight into a radial part (r_t, i) and an angular part (cosθ). While the radial part is well-motivated and ablated, the precise performance contribution of the angular part is less clear. The ablation studies do not isolate its effect. Since the radial component already includes interactions from gating and higher-order softmax terms, it's not immediately obvious how much extra performance is gained by multiplying by the cosθ term, especially when RoPE is already used.

---

> ### Author Rebuttal · Authors · 2025-07-31
>
> ```
> This complexity could make the method harder to analyze...
>
> ... work prioritizes algorithmic expressivity over low-level engineering optimizations ...
> ```
>
> Thank you very much for your recognition of our work, and for understanding our current focus on improving expressivity rather than achieving optimal efficiency. Based on suggestions from other reviewers, we provide here a more detailed discussion and empirical analysis of the computational complexity.
>
> In terms of model complexity, ZeroS can effectively be viewed as an extended form of linear attention, with hidden states of size $3 \times d \times d$. In practice, the implementation only requires maintaining three such hidden state matrices, performing prefix scans on each, and using gated queries to read out the corresponding results, which are then summed to produce the final output. This process is operationally consistent with that of standard linear attention.
>
> In recurrence, ZeroS performs associative scans over hidden states corresponding to 3 different $d \times d$ bases: $e^{s_i} \hat{\mathbf{k}}_i^\top\mathbf{v}_i$, $s_i \hat{\mathbf{k}}_i^\top \mathbf{v}_i$, and $\hat{\mathbf{k}}_i^\top \mathbf{v}_i$. The final output is computed by reading out the scanned results using queries with different gating weights and summing them. Although we have not yet implemented a fused CUDA kernel that performs all scans in a single forward, we can leverage existing linear attention scan implementations (flash linear attention [1]) to perform three separate scans with different key and value bases. This provides a partially efficient implementation:
>
>
> ```python
> # prepare reweighted query q1, q2, and q3 from q...
> out1 = run_linattn(q1, k*s_i_exp, v, mode='fused_chunk')
> out2 = run_linattn(q2, k*s_i, v, mode='fused_chunk')
> out3 = run_linattn(q3, k, v, mode='fused_chunk')
> out = out1 + out2 + out3
> # ...
> ```
>
> To evaluate the runtime efficiency of ZeroS under this implementation, we benchmarked it by replacing the attention layers in a GPT-2 Transformer architecture (dimension = 768, number of heads = 12, sequence length = 1024, number of layers = 12) with various attention variants. We adopted the Transformer and softmax attention implementations (both flash and naive versions) from NanoGPT \[2]. In addition to our partially efficient ZeroS implementation (based on the `fused_chunk_linear_attn` function from flash linear attention \[1]), we also included the following layers provided by the same library as baselines for comparison: \[LinAttn, GatedLinAttn, HGRN2, RWKV6, RWKV7], using their default configurations aside from the GPT-2 settings above. All experiments were conducted on a single Nvidia L40S GPU, using a batch size of 8 and FP32 precision. We measured performance in both training and inference modes (Fwd; full-sequence inference without KV-cache or hidden state cache). After several warm-up forward passes, we recorded the average forward latency, standard error, and peak memory usage using multiple runs synchronized with `torch.cuda.synchronize()`. The results are summarized below.
>
>
> | Model | FwdLatency(s) | FwdStd(s) | TrainLatency(s) | TrainStd(s) | ThroughputFwd(tok/s) | ThroughputTrain(tok/s) | MemPeakFwd(GB) | MemPeakTrain(GB) |
> |---|---|---|---|---|---|---|---|---|
> | SoftMaxAttn (Naïve) | 0.1306 | 0.0006 | 0.3334 | 0.0009 | 62740.89 | 24574.18 | 9.61 | 10.34 |
> | RWKV7 | 0.0876 | 0.0016 | 0.2626 | 0.0010 | 93491.90 | 31199.35 | 9.81 | 10.61 |
> | RWKV6 | 0.0761 | 0.0005 | 0.2252 | 0.0010 | 107653.00 | 36382.92 | 9.49 | 9.62 |
> | ZeroS | 0.0720 | 0.0008 | 0.1974 | 0.0011 | 113855.38 | 41491.29 | 7.48 | 7.61 |
> | HGRN2 | 0.0672 | 0.0013 | 0.1480 | 0.0009 | 121955.16 | 55336.55 | 6.14 | 6.43 |
> | LinAttn | 0.0666 | 0.0010 | 0.1477 | 0.0009 | 122949.71 | 55447.86 | 5.79 | 5.90 |
> | SoftmaxAttn (FlashAttn) | 0.0651 | 0.0014 | 0.1473 | 0.0008 | 125836.28 | 55620.75 | 5.45 | 5.56 |
> | GatedLinAttn | 0.0600 | 0.0009 | 0.1331 | 0.0008 | 136633.08 | 61533.68 | 5.74 | 5.89 |
>
> Compared to other linear attention variants, ZeroS, even without a fully fused associative scan implementation, achieves efficiency on par with the average level of existing architectures, and is not far behind standard linear attention.
>
> ```
> the precise performance contribution of the angular part is less clear.
> ```
>
> Thank you for the valuable question. As shown in Figure 1(c), the Angular (and Value) components align closely with standard linear attention and form the core of effective attention modeling. RoPE and L2 normalization help capture meaningful $\cos \theta_{t,i}$ signals tied to relative position ($t - i$). In contrast, the Radial component is our main contribution, which is why we emphasize it in Figure 1(a,b) and the ablation studies.
>
> To clarify the Angular component’s role, we provide extended ablations analyzing both Angular and Radial parts below.
>
>
> |Model|Compress|FuzzyRecall|In-ContextRecall|Memorize|NoisyRecall|SelectiveCopy|Average|
> |---|---|---|---|---|---|---|---|
> |ZeroS|44.0|14.9|99.9|88.1|96.1|97.8|73.5|
> |w/o Angular|39.5|8.5|42.8|54.5|44.8|63.3|42.2|
> |Ang:w/o PosEmb|35.8|9.4|73.3|46.2|66.2|45.8|46.1|
> |Ang:w/ additive PosEmb|38.1|14.2|94.1|86.6|87.2|93.8|69.0|
> |w/o Radial|35.9|9.6|84.8|86.3|86.5|92.3|65.9|
> |Rad:standard SM|36.3|10.6|91.8|81.7|89.7|95.3|67.6|
> |Rad:Gated ZeroS but add 0-th back|42.0|10.5|91.4|85.2|90.0|97.1|69.4|
> |Rad:Zeros w/o Gating|39.7|13.5|96.3|83.0|94.6|97.8|70.8|
>
>
> Removing the Angular part (w/o Angular) eliminates the bilinear query-key interactions ($x_t \widetilde{\mathbf{W}}_q \widetilde{\mathbf{W}}_k x_i$), leaving only scalar weight products between steps $t$ and $i$. This reduces the model to a $3 \times d$ hidden-state RNN scan over values, instead of the full $3 \times d \times d$ structure, significantly lowering expressiveness.
>
> Reintroducing queries and keys without positional embeddings (Ang: w/o PosEmb) offers only a minor gain, as $\cos \theta_{t,i}$ lacks positional meaning. Using additive positional embeddings (Ang: w/ additive PosEmb) improves performance, comparable to strong baselines—but still below ZeroS with RoPE. This highlights the importance of modeling relative positions via $\cos \theta_{t,i}$, aligning with prior findings on linear attention.
>
> ```
> a sequence-to-sequence task that standard attention would struggle with but that zero-sum attention can solve easily in a single layer...
> ```
>
> Thank you for the insightful question. Standard softmax attention restricts outputs to the convex hull of input values, $\mathrm{Conv}({\mathbf{v}_i})$, allowing only non-negative weights. In contrast, ZeroS enables zero-sum combinations, $\{\sum_i w_i \mathbf{v}_i \mid \sum_i w_i = 0\}$, allowing it to express operations like differencing and contrast, crucial for many reasoning tasks.
>
> This expanded expressiveness comes with proven numerical stability, and is further enhanced by gating for higher-order terms. While discussed limitations of convex weights are not new, our work (see Section 2.2) builds on this to propose ZeroS as an effective and stable solution.
>
> To illustrate this, we provide examples of differential and contrast tasks (see refs [27, 41–43] of main paper) that ZeroS naturally supports.
>
> - **1. Minimal Counterexample: Two-Token Difference**
>
> |**Task Definition**|**Output:** $\mathbf{v}_1-\mathbf{v}_2$|
> |---|---|
> |**Standard Softmax**|$\mathbf{o}=\alpha_1\mathbf{v}_1+\alpha_2\mathbf{v}_2,\ \alpha_1,\alpha_2\ge0,\ \alpha_1+\alpha_2=1$|
> |**Constraint**|$\Rightarrow\alpha_1=1,\ \alpha_2=-1$ violates non-negativity|
> |**ZeroS Attention**|$\mathbf{o}=w_1\mathbf{v}_1+w_2\mathbf{v}_2,\ w_1+w_2=0,\ w_1=1,\ w_2=-1\Rightarrow\mathbf{o}=\mathbf{v}_1-\mathbf{v}_2$|
>
> - **2. Two Simple Task Examples**
>
> |**Task**|**Softmax Fail**|**ZeroS Can Achieve**|
> |---|---|---|
> |(a) Difference Between a Token and Mean|$\mathbf{o}=\mathbf{v}\_1-\frac{1}{t}\sum_{i=1}^t\mathbf{v}\_i$, requires neg weights|$w_1=1-\frac{1}{t},\ w_{i>1}=-\frac{1}{t},\ \sum w_i=0$|
> |(b) Alternating Difference Pattern|$\mathbf{o}=\sum\_{i=1}^{t/2}(\mathbf{v}\_{2i-1}-\mathbf{v}\_{2i})$, requires alternating ± weights|$w_{2i-1}=1,\ w_{2i}=-1,\ \sum w_i=0$|
>
>
> ```
> Do you see any downsides or trade-offs to applying this modification to standard softmax attention in models where quadratic complexity is not a bottleneck?
> ```
>
> Thank you for the thoughtful question. The trade-offs of applying ZeroS to softmax attention are similar to those seen in the linear case.
>
> In Table 1, ZeroS (Lin) outperforms standard linear attention across all tasks due to the weaker baseline. For ZeroS (SM) vs. softmax, most gains appear in *In-Context Recall and Noisy Recall*, which require precise retrieval and robustness, reflecting key challenges in complex sequence modeling such as precise retrieval of prior information and robustness to noise. This supports our view that ZeroS improves expressiveness for harder tasks.
>
> In contrast, for tasks like Fuzzy Recall that favor simple averaging, ZeroS may slightly underperform, as it lacks the softmax’s built-in averaging bias. Learning such behavior from scratch may require more layers or parameters, though averaging itself is cheap and doesn't pose a major bottleneck.
>
> If a task’s complexity or bias is unclear (particularly whether it relies on simple averaging), we recommend using a multi-layer ZeroS as a general-purpose attention architecture.
>
> ### References
>
> [1] Yang, S., & Zhang, Y. (2024). Fla: A triton-based library for hardware-efficient implementations of linear attention mechanism (https://github.com/fla-org/flash-linear-attention)
>
> [2] Karpathy, A. (2022). NanoGPT. GitHub Repository. https://github.com/karpathy/nanoGPT

---

### Official Review · Reviewer_kvKy · 2025-07-05

**Clarity:** 2
**Significance:** 2
**Originality:** 3
**Rating:** 4
**Confidence:** 4

**Summary:**

The paper addresses the performance gap between efficient linear attention models and standard quadratic softmax attention, which is attributed to the positive convex weights that makes subtraction impossible. The authors propose a zero-sum attention mechanism so that the weights can be both positive and negative, which leads to both quadratic and linear attention variants. Experiments in language modeling, in-context learning, and time series forecasting show that ZeroS matches or exceeds the performance of baselines.

**Questions:**

The reviewer would love to see the concerns in the above addressed and revise the rating.

**Ethical Concerns:**

["NO or VERY MINOR ethics concerns only"]

**Final Justification:**

I am happy with the experiments on space and time complexity, as well as the explanation of efficient forward pass. On the other hand, based on the authors response, I am concerned about the impact of experimenting on a benchmark with context length smaller than 4K: modern applications requires LLMs to reason with a large context length, and this is the regime where a method being both effective and efficient can make a difference. Therefore, I maintain my original rating.

**Limitations:**

yes

**Quality:**

3

**Strengths And Weaknesses:**

## Significance

1. Metrics: A major metric that is missing from the evaluation is the wall-clock running time.
- On the one hand, I understand that the asymptotic complexities have been provided.
- On the other hand, there are aspects not captured by such complexities, including how friendly the computations are to caching, the constants in the asymptotics which matter before sufficiently large N and d are reached, etc.
- It might be demanding to do it for all baselines, but at least comparing representative linear/quadratic baselines as well as variants in your ablation study can give a better picture. For instance, MultiHead Hyena, DeltaNet, and Transformer \- how do their running time compare with that of ZeroS variants?
2. Baselines: Since efficient transformer architecture is a fast paced direction, it helps make an impact by comparing with recent open-source baselines
- (ICLR 24\) https://openreview.net/forum?id=Eh0Od2BJIM
- (ICLR 25\) https://openreview.net/forum?id=lXRDQsiP2v

## Clarity

1. Figure 1 is a bit complicated and hard to parse. For a novice reader that scans the figures before reading the paper, it is hard to appreciate the method. Specifically:
- There is no label of {u\_i} in figure 1 or its caption. One has to refer to the text.
- The authors may consider contrasting the proposed architecture with a representative baseline, so that it is immediately clear which parts are similar, and which parts are different (and novel).
2. Figure 2 has very small labels and legends making it difficult to read.
3. Line 134-136 has little context, since the proposed approach has not been introduced yet.
4. Typos:
- Line 146: something is missing after “∑ i w\_i=”
- The equation block below line 207 has duplications

---

> ### Author Rebuttal · Authors · 2025-07-31
>
> ```
> Metrics: A major metric that is missing from the evaluation is the wall-clock running time.
> ```
>
> Thank you very much for your valuable question and suggestion. This is extremely helpful for users interested in implementing ZeroS in practice. We will include a concise version of the following explanation in the main paper and provide a more detailed version in the appendix during the revision.
>
>
> In recurrence, ZeroS performs associative scans over hidden states corresponding to 3 different $d \times d$ bases: $e^{s_i} \hat{\mathbf{k}}_i^\top\mathbf{v}_i$, $s_i \hat{\mathbf{k}}_i^\top \mathbf{v}_i$, and $\hat{\mathbf{k}}_i^\top \mathbf{v}_i$. The final output is computed by reading out the scanned results using queries with different gating weights and summing them. Although we have not yet implemented a fused CUDA kernel that performs all scans in a single forward, we can leverage existing linear attention scan implementations (flash linear attention [1]) to perform three separate scans with different key and value bases. This provides a partially efficient implementation:
>
>
> ```python
> # prepare reweighted query q1, q2, and q3 from q...
> out1 = run_linattn(q1, k*s_i_exp, v, mode='fused_chunk')
> out2 = run_linattn(q2, k*s_i, v, mode='fused_chunk')
> out3 = run_linattn(q3, k, v, mode='fused_chunk')
> out = out1 + out2 + out3
> # ...
> ```
>
> To evaluate the runtime efficiency of ZeroS under this implementation, we benchmarked it by replacing the attention layers in a GPT-2 Transformer architecture (dimension = 768, number of heads = 12, sequence length = 1024, number of layers = 12) with various attention variants. We adopted the Transformer and softmax attention implementations (both flash and naive versions) from NanoGPT \[2]. In addition to our partially efficient ZeroS implementation (based on the `fused_chunk_linear_attn` function from flash linear attention \[1]), we also included the following layers provided by the same library as baselines for comparison: \[LinAttn, GatedLinAttn, HGRN2, RWKV6, RWKV7], using their default configurations aside from the GPT-2 settings above. All experiments were conducted on a single Nvidia L40S GPU, using a batch size of 8 and FP32 precision. We measured performance in both training and inference modes (Fwd; full-sequence inference without KV-cache or hidden state cache). After several warm-up forward passes, we recorded the average forward latency, standard error, and peak memory usage using multiple runs synchronized with `torch.cuda.synchronize()`. The results are summarized below.
>
>
> | Model | FwdLatency(s) | FwdStd(s) | TrainLatency(s) | TrainStd(s) | ThroughputFwd(tok/s) | ThroughputTrain(tok/s) | MemPeakFwd(GB) | MemPeakTrain(GB) |
> |---|---|---|---|---|---|---|---|---|
> | SoftMaxAttn (Naïve) | 0.1306 | 0.0006 | 0.3334 | 0.0009 | 62740.89 | 24574.18 | 9.61 | 10.34 |
> | RWKV7 | 0.0876 | 0.0016 | 0.2626 | 0.0010 | 93491.90 | 31199.35 | 9.81 | 10.61 |
> | RWKV6 | 0.0761 | 0.0005 | 0.2252 | 0.0010 | 107653.00 | 36382.92 | 9.49 | 9.62 |
> | ZeroS | 0.0720 | 0.0008 | 0.1974 | 0.0011 | 113855.38 | 41491.29 | 7.48 | 7.61 |
> | HGRN2 | 0.0672 | 0.0013 | 0.1480 | 0.0009 | 121955.16 | 55336.55 | 6.14 | 6.43 |
> | LinAttn | 0.0666 | 0.0010 | 0.1477 | 0.0009 | 122949.71 | 55447.86 | 5.79 | 5.90 |
> | SoftmaxAttn (FlashAttn) | 0.0651 | 0.0014 | 0.1473 | 0.0008 | 125836.28 | 55620.75 | 5.45 | 5.56 |
> | GatedLinAttn | 0.0600 | 0.0009 | 0.1331 | 0.0008 | 136633.08 | 61533.68 | 5.74 | 5.89 |
>
> Compared to other linear attention variants, ZeroS, even without a fully fused associative scan implementation, achieves efficiency on par with the average level of existing architectures, and is not far behind standard linear attention.
>
> ```
> About more baselines...
> ```
>
> Thank you very much for your helpful suggestions regarding the experimental section. Taking into account feedback from all reviewers, we have added several recently developed attention variants as additional baselines for comparison against ZeroS on the MAD benchmark. These include: [HedgeDog, Hawk, Mamba2, xLSTM, Gated DeltaNet].
>
> Regarding the two baselines you specifically mentioned:
>
> 1. For HyperAttention, according to its official implementation (https://github.com/insuhan/hyper-attn), the default value for `min_seq_len` (which determines when hyper attention is activated) is set to 4096. If the input sequence length is below this threshold, the model simply defaults to flash attention instead of using hyper attention. Since the sequence lengths in our benchmark tasks are generally below this threshold, the behavior of hyper attention essentially mirrors that of softmax attention in this setting. Therefore, you may interpret the softmax attention results in our baselines as a proxy for HyperAttention performance under these conditions.
>
> 2. For ToST [3], the main focus of the paper is on non-causal sequence modeling tasks such as visual tasks and LRA. Causal modeling is only briefly discussed as an extension, and there is no comparison against other models in that context. In contrast, our paper focuses primarily on causal tasks, with the MAD benchmark mainly about causal scenarios. To avoid biased comparisons across different task scopes, we followed the precedent set by other recent works in selecting baselines and did not include ToST in our experiments.
>
> That said, we will expand the discussion in the related works section of our revision to better contextualize and contrast attention variants designed for different modeling purposes. Thank you again for the suggestion.
>
> We present the extended comparison with newly added baselines as follows:
>
> | Model | Compress | Fuzzy Recall | In-Context Recall | Memorize | Noisy Recall | Selective Copy | Average |
> |---|---|---|---|---|---|---|---|
> | ZeroS (Softmax Attn) | 45.2 | 28.0 | 100.0 | 84.3 | 96.6 | 98.5 | 75.4 |
> | Softmax Attn | 51.6 | 29.8 | 94.1 | 85.2 | 86.8 | 99.6 | 74.5 |
> | ZeroS (Lin) | 44.0 | 14.9 | 99.9 | 88.1 | 96.1 | 97.8 | 73.5 |
> | xLSTM | 43.4 | 47.6 | 100.0 | 79.8 | 100.0 | 95.4 | 73.2 |
> | Mamba2 | 43.6 | 21.1 | 96.4 | 86.9 | 96.7 | 93.3 | 73.0 |
> | GDeltaNet | 45.0 | 32.7 | 100.0 | 81.7 | 100.0 | 95.7 | 71.0 |
> | Hawk | 47.7 | 13.6 | 93.0 | 91.3 | 93.0 | 77.0 | 64.5 |
> | LinAttn | 33.1 | 8.2 | 91.0 | 74.9 | 75.6 | 93.1 | 62.3 |
> | HedgeDog | 43.2 | 17.9 | 55.9 | 83.4 | 46.0 | 98.4 | 57.4 |
>
> As shown, ZeroS maintains a consistent advantage in average performance across a variety of task types compared to recent baselines. Further discussion can be found in our response to Reviewer FheY.
>
> ```
> Figure 1 is a bit complicated and hard to parse...
> ```
>
> Thank you very much for your valuable comments and suggestions: they are extremely helpful in improving the clarity and readability of our paper. We will address these issues in the revision.
>
> For the notation $\{u_i\}$, we will add a clearer annotation indicating that it is obtained by applying a linear projection layer $\mathbf{W}_u$ to the input $\{x_i\}$.
>
> Regarding the comment "contrasting the proposed architecture with a representative baseline...", we will revise Figure 1(c) to explicitly label the *Angular* and *Value* components as common elements found in existing linear attention variants, while highlighting the *Radial* component as the main focus and novel contribution of our work.
>
> ```
> Figure 2 has very small labels and legends making it difficult to read.
> ```
>
> Thank you for pointing this out. We will redraw Figure 2 to improve readability by enlarging the legends and labels.
>
> ```
> Line 134-136 has little context...
> ```
>
> Thank you for the suggestion. We will revise this section to ensure that the discussion is clearly contextualized within the topic of the softmax function. Any extended discussion about its application in attention will be moved to a more appropriate part of the paper.
>
> ```
> Line 146: something is missing after “∑ i w_i=”
> ```
>
> We greatly appreciate your close reading. Apologies for this oversight: it was caused by compression and restructuring during final editing of the mathematical sections. The right-hand side of the equation at Line 146 is indeed missing the zero-sum constraint, i.e., it should be $ \sum_i w_i = 0 $. We will correct this in the revision. Additionally, in Line 207, we will retain the second line and remove the preceding portion for clarity.
>
>
> ### References
>
> [1] Yang, S., & Zhang, Y. (2024). Fla: A triton-based library for hardware-efficient implementations of linear attention mechanism (https://github.com/fla-org/flash-linear-attention)
>
> [2] Karpathy, A. (2022). NanoGPT. GitHub Repository. https://github.com/karpathy/nanoGPT
>
> [3] Wu, Z., Ding, T., Lu, Y., Pai, D., Zhang, J., Wang, W., ... & Haeffele, B. D. Token Statistics Transformer: Linear-Time Attention via Variational Rate Reduction. In The Thirteenth International Conference on Learning Representations.

---

> > ### Comment · Reviewer_kvKy · 2025-08-07
> > **Thanks to the authors for their response**
> >
> > Thank you for your response!
> >
> > I am happy with the experiments on space and time complexity, as well as the explanation of efficient forward pass. On the other hand, based on the authors response, I am concerned about the impact of experimenting on a benchmark with context length smaller than 4K: modern applications requires LLMs to reason with a large context length, and this is the regime where a method being both effective and efficient can make a difference.
> >
> > I have read authors's reply to other reviewers. Overall, I am leaning towards the paper being accepted.

---

> > > ### Author Response · Authors · 2025-08-07
> > >
> > > Thank you very much for your recognition and positive assessment of our work. As stated in the limitations section of our paper, our focus is primarily on theoretical algorithmic improvements and enhanced expressivity. To support this, we evaluate on a set of recently adopted, fine-grained tasks to illustrate which types of tasks benefit most from the expressivity gains. As noted in the paper, on the engineering side, we have not yet implemented fused propagation for optimal efficiency, and we have not yet explored scaling up to larger model sizes.
> > >
> > > That said, we would like to highlight that our method offers a theoretically stronger stability bound over classic linear attention at longer sequence lengths $t$, leading to improved numerical stability for long sequence modeling (see Section 3.2 and our discussion with Reviewer FheY regarding normalization terms). Moreover, all of the synthetic subtasks we evaluate on include varying sequence lengths, minimizing bias from sequence length differences within the available resources. ZeroS shows performance gains over existing linear attention methods across varying lengths in our experiments.
> > >
> > > We hope this clarifies the intended scope of our work and the balance we chose between theoretical insights and engineering implementation.
> > >
> > > Once again, thank you for your thoughtful feedback and recognition of our work.

---

### Official Review · Reviewer_WRY9 · 2025-07-10

**Clarity:** 3
**Significance:** 4
**Originality:** 4
**Rating:** 5
**Confidence:** 5

**Summary:**

This work identifies key issues in the Linear Attention method proposed in [1] and introduces Zero-Sum Linear Attention to address them. First, it argues that the zeroth-order term in attention dilutes the weights. Removing this term not only resolves that issue but also permits negative weights—naturally contrastive—building on insights from Differential Attention [2]. The authors further show that zero-sum weights can improve expressivity over the convex combination in standard attention and that the zero-sum form is stable under bounded logits. They then discuss the benefits of decoupling direction and magnitude. Finally, they demonstrate that the inherent negative weights and stability of zero-sum attention let them drop the angular term ($\cos\theta$) from the denominator, seamlessly integrating with RoPE as a by-product. Extensive experiments across diverse benchmarks and domains indicate that Zero-Sum Linear Attention matches standard attention closely while outperforming other linear variants.

References:

[1] Fast Autoregressive Transformers with Linear Attention

[2] Differential Transformer

**Questions:**

**Questions and Suggestion**

- Line 34: “In long contexts, attention mechanisms incorporate a roughly uniform component … introducing a persistent averaging effect …”

For standard softmax attention, large query–key dot products often make the distribution very peaky. emperically the attention patterns also tend to be sparse in the upper layers, so the averaging effect may be less pronounced than implied. For linear attention though, the averaging effect tends to be more pronounced.

- Line 98.: “As weights become more uniform …”
  Assuming this about attention weights, i would suggest to make this explicit.

- Line 227. In the linear recurrent equations, the zeroth-order term appears to be absent. It would help to state this explicitly to remind the readers.

- A short discussion for the non-causal variant would help improve the clarity for the paper.

- Table 1. Any hypothesis for the weaker performance on the Fuzzy Recall task?

- To keep the paper self-contained, please add brief descriptions of the benchmarks—particularly RegBench and MAD in appendix.

**Ethical Concerns:**

["NO or VERY MINOR ethics concerns only"]

**Final Justification:**

I have read the reviews and rebuttal and am satisfied with the response. Thus I have kept my original rating of acceptance

**Quality:**

3

**Strengths And Weaknesses:**

**Strengths**

- The paper proposes several novel ideas:
  - removing the zeroth-order term,
  - decoupling angular and radial components, and
  - using effective gating plus higher-order terms via differentiation—together yielding improved accuracy for linear-attention methods.
- The manuscript is generally clear, with only minor writing issues.
- The contributions are well motivated, supported by sound theoretical arguments and strong empirical results.

**Weaknesses**
- The major weakness is the absence of an efficient implementation for the recurrent equations. Although the authors note this in the “Limitations” section, it remains a significant limitation for practical adoption. At minimum, please report time- and memory-usage comparisons against standard attention implementations such as FlashAttention-2.

- Typos / notation:
  - Line 146: “$\sum_i w_i$ = …” (missing RHS).
  - Line 196: “output be$o_t$” → “output be $o_t$”.
  - Equation above line 208: should the index be $\varepsilon_{ti}$, not $\varepsilon_{i}$?
  - Line 208: should this be $\sigma_t^1 = \operatorname{sigmoid}(x_t W_g^1)$ ? and similarly for $\sigma_t^h$.

---

> ### Author Rebuttal · Authors · 2025-07-31
>
> ```
> the absence of an efficient implementation for the recurrent equations...
> ```
>
> Thank you very much for your recognition of our paper and for your insightful questions. We will include a concise version of the following explanation in the main paper and provide an extended version in the appendix during the revision.
>
> In recurrence, ZeroS performs associative scans over hidden states corresponding to 3 different $d \times d$ bases: $e^{s_i} \hat{\mathbf{k}}_i^\top\mathbf{v}_i$, $s_i \hat{\mathbf{k}}_i^\top \mathbf{v}_i$, and $\hat{\mathbf{k}}_i^\top \mathbf{v}_i$. The final output is computed by reading out the scanned results using queries with different gating weights and summing them. Although we have not yet implemented a fused CUDA kernel that performs all scans in a single forward, we can leverage existing linear attention scan implementations (flash linear attention [1]) to perform three separate scans with different key and value bases. This provides a partially efficient implementation:
>
>
> ```python
> # prepare reweighted query q1, q2, and q3 from q...
> out1 = run_linattn(q1, k*s_i_exp, v, mode='fused_chunk')
> out2 = run_linattn(q2, k*s_i, v, mode='fused_chunk')
> out3 = run_linattn(q3, k, v, mode='fused_chunk')
> out = out1 + out2 + out3
> # ...
> ```
>
> To evaluate the runtime efficiency of ZeroS under this implementation, we benchmarked it by replacing the attention layers in a GPT-2 Transformer architecture (dimension = 768, number of heads = 12, sequence length = 1024, number of layers = 12) with various attention variants. We adopted the Transformer and softmax attention implementations (both flash and naive versions) from NanoGPT \[2]. In addition to our partially efficient ZeroS implementation (based on the `fused_chunk_linear_attn` function from flash linear attention \[1]), we also included the following layers provided by the same library as baselines for comparison: \[LinAttn, GatedLinAttn, HGRN2, RWKV6, RWKV7], using their default configurations aside from the GPT-2 settings above. All experiments were conducted on a single Nvidia L40S GPU, using a batch size of 8 and FP32 precision. We measured performance in both training and inference modes (Fwd; full-sequence inference without KV-cache or hidden state cache). After several warm-up forward passes, we recorded the average forward latency, standard error, and peak memory usage using multiple runs synchronized with `torch.cuda.synchronize()`. The results are summarized below.
>
>
> | Model | FwdLatency(s) | FwdStd(s) | TrainLatency(s) | TrainStd(s) | ThroughputFwd(tok/s) | ThroughputTrain(tok/s) | MemPeakFwd(GB) | MemPeakTrain(GB) |
> |---|---|---|---|---|---|---|---|---|
> | SoftMaxAttn (Naïve) | 0.1306 | 0.0006 | 0.3334 | 0.0009 | 62740.89 | 24574.18 | 9.61 | 10.34 |
> | RWKV7 | 0.0876 | 0.0016 | 0.2626 | 0.0010 | 93491.90 | 31199.35 | 9.81 | 10.61 |
> | RWKV6 | 0.0761 | 0.0005 | 0.2252 | 0.0010 | 107653.00 | 36382.92 | 9.49 | 9.62 |
> | ZeroS | 0.0720 | 0.0008 | 0.1974 | 0.0011 | 113855.38 | 41491.29 | 7.48 | 7.61 |
> | HGRN2 | 0.0672 | 0.0013 | 0.1480 | 0.0009 | 121955.16 | 55336.55 | 6.14 | 6.43 |
> | LinAttn | 0.0666 | 0.0010 | 0.1477 | 0.0009 | 122949.71 | 55447.86 | 5.79 | 5.90 |
> | SoftmaxAttn (FlashAttn) | 0.0651 | 0.0014 | 0.1473 | 0.0008 | 125836.28 | 55620.75 | 5.45 | 5.56 |
> | GatedLinAttn | 0.0600 | 0.0009 | 0.1331 | 0.0008 | 136633.08 | 61533.68 | 5.74 | 5.89 |
>
> Compared to other linear attention variants, ZeroS, even without a fully fused associative scan implementation, achieves efficiency on par with the average level of existing architectures, and is not far behind standard linear attention.
>
> ```
> Typos / notation:
> ```
>
> Thank you very much for your careful review: your feedback is very helpful in improving the quality of our paper. We sincerely apologize for the errors introduced due to space reduction in the theory section during final editing.
>
> * In Line 146, the RHS is missing the zero-sum weights condition, i.e., it should include $= 0$.
> * Line 196 will have a better formatting.
> * For Line 208, we apologize for the confusion. It should read: $\sigma\_t^1 = \text{sigmoid}(\mathbf{x}\_t \mathbf{W}\_{g}^1), \quad \sigma\_t^h = \text{sigmoid}(\mathbf{x}\_t \mathbf{W}\_{g}^h)$, indicating that the gating weights are determined by the input information at step \$t\$.
>
>
> ```
> Line 34: “In long contexts, attention mechanisms incorporate a roughly uniform component … introducing a persistent averaging effect …”
> ```
>
> We apologize for the confusion. Your interpretation aligns with our intended meaning. Our logic here is that the expressiveness of softmax attention arises from its ability to deviate from the average distribution (e.g., produce peaked patterns). However, softmax inherently includes a zero-order averaging effect, approximately $1/t$, which is input-independent. The higher-order deviations are responsible for offsetting this persistent averaging and producing more expressive distributions like sharp peaks.
>
> By removing this zero-order constant normalization, we can free the higher-order terms from the burden of canceling out the averaging effect. This allows them to more efficiently model peaked distributions aligned in QK interactions. Furthermore, we show that eliminating the zero-order term (via zero-sum attention weights) does not lead to numerical instability as sequence length increases, while it enhances expressivity by enabling modeling capabilities, such as differential and contrast patterns that a single softmax attention layer cannot represent. We believe this makes zero-sum attention a more effective form.
>
> We will revise Section 3.1 to better clarify this point.
>
> ```
> Line 98.: “As weights become more uniform …” Assuming this about attention weights
> ```
>
> We will revise the sentence to explicitly include the term "attention weights."
>
> ```
> Line 227. ...the zeroth-order term appears to be absent. It would help to state this explicitly to remind the readers.
> ```
>
> Thank you for the suggestion. We will explicitly state around Line 227 that the zero-order term has a fixed gating factor of zero. This makes clear that, unlike higher-order terms with trainable gating, the zero-order normalization is intentionally disabled.
>
> ```
> A short discussion for the non-causal variant would help improve the clarity for the paper.
> ```
>
> Thank you for the suggestion. We will add a discussion on this in the appendix. Similar to prior work on linear attention, our method can be made causal or non-causal depending on how the KV hidden states are handled: using a prefix scan results in a causal variant, while applying a average-pooling hidden state across all output steps yields the non-causal version.
>
> ```
> Table 1. Any hypothesis for the weaker performance on the Fuzzy Recall task?
> ```
>
> Thank you for this insightful question. As shown in Table 1, ZeroS shows the most significant improvements over linear and softmax attention on the *In-Context Recall* and *Noisy Recall* tasks. These tasks focus on accurately retrieving specific earlier information from complex sequences (a core challenge in language modeling). The observed performance gains align with our motivation for ZeroS, which is to enhance the expressiveness of higher-order deviations beyond the averaging effects of standard attention.
>
> By contrast, the *Fuzzy Recall* task is more like soft averaging over the input context, which can be effectively handled by simpler, lower-cost structures. Modeling precise alignment is less critical here, and models that naturally perform soft averaging may have a structural bias advantage. Even so, ZeroS still shows a clear improvement over the original linear attention baseline, and only slightly underperforms softmax attention, likely because it lacks the same inductive bias toward averaging input tokens, which the model must instead learn explicitly. Further discussion can be found in our response to Reviewer FheY.
>
> ```
> To keep the paper self-contained, please add brief descriptions of the benchmarks—particularly RegBench and MAD in appendix.
> ```
>
> Thank you for the suggestion. We will include more task-specific descriptions of RegBench and MAD in Appendix A.3 to make the paper more self-contained.
>
> ### References
>
> [1] Yang, S., & Zhang, Y. (2024). Fla: A triton-based library for hardware-efficient implementations of linear attention mechanism (https://github.com/fla-org/flash-linear-attention)
>
> [2] Karpathy, A. (2022). NanoGPT. GitHub Repository. https://github.com/karpathy/nanoGPT

---

### Decision · Program_Chairs · 2025-09-17

**Decision:**

Accept (spotlight)

**Comment:**

This paper proposes an improvement to linear attention stemming from a careful analysis of kernel approximation of softmax attention. This results in several modifications, including the removal of the zeroth order term, residual reweighting, and decoupling of the angular/radial components. Empirical experiments demonstrate the effectiveness of this approach across multiple benchmarks, including diagnostic benchmarks (MAD/Regbench) as real-world domains (language/image/time-series modeling).

All reviewers appreciated the fact that the method was well-motivated and moreover backed up by both theory and solid empirical results spanning diverse domains. It is also noteworthy that the approach can also improve softmax attention layers. There were some initial concerns with regard to wallclock efficiency, but these issues were largely addressed during the rebuttal.

The main weakness is scale: language modeling experiments are only conducted at very small scales. However, I think this is ok given the significant experiments in other domains.

Overall, I believe that this work represents an excellent contribution to the architecture community and deserves to be highlighted at the conference.